# Stochastic Taylor Derivative Estimator: Efficient amortization for arbitrary differential operators

**Zekun Shi**
National University of Singapore
Sea AI Lab
shizk@sea.com,

**Zheyuan Hu**
National University of Singapore
e0792494@u.nus.edu,

**Min Lin**
Sea AI Lab
linmin@sea.com,

**Kenji Kawaguchi**
National University of Singapore
kenji@nus.edu.sg

## Abstract

Optimizing neural networks with loss that contain high-dimensional and high-order differential operators is expensive to evaluate with back-propagation due to $\mathcal{O}(d^k)$ scaling of the derivative tensor size and the $\mathcal{O}(2^{k-1}L)$ scaling in the computation graph, where $d$ is the dimension of the domain, $L$ is the number of ops in the forward computation graph, and $k$ is the derivative order. In previous works, the polynomial scaling in $d$ was addressed by amortizing the computation over the optimization process via randomization. Separately, the exponential scaling in $k$ for univariate functions ($d = 1$) was addressed with high-order auto-differentiation (AD). In this work, we show how to efficiently perform arbitrary contraction of the derivative tensor of arbitrary order for multivariate functions, by properly constructing the input tangents to univariate high-order AD, which can be used to efficiently randomize any differential operator. When applied to Physics-Informed Neural Networks (PINNs), our method provides >1000× speed-up and >30× memory reduction over randomization with first-order AD, and we can now solve *1-million-dimensional PDEs in 8 minutes on a single NVIDIA A100 GPU*[1]. This work opens the possibility of using high-order differential operators in large-scale problems.

## 1 Introduction

In many problems, especially in Physics-informed machine learning [19, 33], one needs to solve optimization problems where the loss contains differential operators:

$$\arg\min_{\theta} f(\mathbf{x}, u_\theta(\mathbf{x}), \mathcal{D}^{\alpha^{(1)}} u_\theta(\mathbf{x}), \dots, \mathcal{D}^{\alpha^{(n)}} u_\theta(\mathbf{x})), \quad u_\theta : \mathbb{R}^d \to \mathbb{R}^{d'}. \tag{1}$$

In this above, $\mathcal{D}^\alpha = \frac{\partial^{|\alpha|}}{\partial x_1^{\alpha_1}, \dots, \partial x_d^{\alpha_d}}$, $\alpha = (\alpha_1, \alpha_2, \dots, \alpha_d)$ is a multi-index, $u_\theta$ is some neural network parameterized by $\theta$, and $f$ is some cost function. When either the differentiation order $k$ or the dimensionality $d$ is high, the objective function above is expensive to evaluate with back-propagation (backward mode AD) in both memory and computation: the size of the derivative tensor has scaling $\mathcal{O}(d^k)$, and the size of the computation graph has scaling $\mathcal{O}(2^{k-1}L)$ where $L$ is the number of ops in the forward computation graph.

---

[1] Our code is available at `https://github.com/sail-sg/stde`
Received the Best Paper Award at NeurIPS 2024

38th Conference on Neural Information Processing Systems (NeurIPS 2024).

There have been several efforts to tackle this curse of dimensionality. One line of work uses randomization to amortize the cost of computing differential operators with AD over the optimization process so that the $d$ in the above scaling becomes a constant for the case of $k = 2$. Stochastic Dimension Gradient Descent (SDGD) [13] randomizes over the input dimensions where in each iteration, the partial derivatives are only calculated for a minibatch of sampled dimensions with back-propagation. In [12, 21, 15], the classical technique of Hutchinson Trace Estimator (HTE) [16] is used to estimate the trace of Hessian or Jacobian to inputs. Others choose to bypass AD completely to reduce the complexity of computation. In [31], the finite difference method is used for estimating the Hessian trace. Randomized smoothing [11, 14] uses the expectation over Gaussian random variable as ansatz, so that its derivatives can be expressed as another expectation Gaussian random variable via Stein's identity [39]. However, compared to AD, the accuracy of these methods is highly dependent on the choice of discretization.

In this work, we address the scaling issue in both $d$ and $k$ for the optimization problem in Eq. 1 at the same time, by proposing an amortization scheme that can be efficiently evaluated via high-order AD, which we call *Stochastic Taylor Derivative Estimator (STDE)*. Our **main contributions** are:

- We demonstrate how Taylor mode AD [6], a high-order AD method, can be used to amortize the optimization problem in Eq. 1. Specifically, we show that, with properly constructed input tangents, the univariate Taylor mode can be used to contract multivariate functions' derivative tensor of arbitrary order;

- We provide a comprehensive procedure for randomizing arbitrary differential operators with STDE, while previous works mainly focus on the Laplacian operator, and we provide abundant examples of STDE constructed for operators in common PDEs;

- STDE encompass and generalizes previous methods like SDGD [13] and HTE [16, 12]. We also prove that HTE-type estimator cannot be generalized beyond fourth order differential operator;

- We determine the efficacy of STDE experimentally. When applied to PINN, our method provides a significant speed-up compared to the baseline method SDGD [13] and the backward-free method like random smoothing [11]. Due to STDE's low memory requirements and reduced computation complexity, PINNs with STDE can **solve 1-million-dimensional PDEs on a single NVIDIA A100 40GB GPU within 8 minutes**, which shows that PINNs have the potential to solve complex real-world problems that can be modeled as high-dimensional PDEs. We also provide a detailed ablation study on the source of performance gain of our method.

## 2    Related works

**High-order and forward mode AD**    The idea of generalizing forward mode AD to high-order derivatives has existed in the AD community for a long time [5, 18, 40, 22]. However, accessible implementation for machine learning was not available until the recent implementation in JAX [6, 7], which implemented the Taylor mode AD for accelerating ODE solver. There are also efforts in creating the forward rule for a specific operator like the Laplacian [24, 23]. Randomization over the linearized part of the AD computation graph was considered in [30]. Forward mode AD can also be used to compute neural network parameter gradient as shown in [2].

**Randomized Gradient Estimation**    Randomization [28, 29, 8] is a common technique for tackling the curse of dimensionality for numerical linear algebra computation, which can be applied naturally in amortized optimization [1]. Hutchinson trace estimator [16] is a well-known technique, which has been applied to diffusion model [37] and PINNs [12]. Another case that requires gradient estimation is when the analytical form of the target function is not available (black box), which means AD cannot be applied. The method of zeroth-order optimization [25] can be used in this case, as it only requires evaluating the function at arbitrary input. It is also useful when the function is very complicated like in the case of a large language model [27].

## 3    Preliminaries and discussions

### 3.1    First-order auto-differentiation (AD)

AD is a technique for evaluating the gradient of composition of known analytical functions commonly called primitives. In an AD framework, a neural network $F_\theta : \mathbb{R}^d \to \mathbb{R}^{d'}$ is constructed as the composition of primitives $F_i$ that are parameterized by some parameters $\theta_i$. In this section, we will

consider the neural networks with linear computation graphs like $F = F_L \circ F_{L-1} \circ \cdots \circ F_1$, but the results generalize to arbitrary directed acyclic graphs (DAGs). We will assume that all hidden dimensions are $h$. See Appendix B for more details on first-order AD.

**Forward mode AD**   Each primitives $F_i$ is linearized as the Fréchet (directional) derivative $\partial F_i$ : $\mathbb{R}^h \to \mathrm{L}(\mathbb{R}^h, \mathbb{R}^h)$, which computes the Jacobian-vector-product (JVP): $\partial F_i(\mathbf{a})(\mathbf{v}) = \frac{\partial F}{\partial \mathbf{x}}\big|_{\mathbf{a}} \mathbf{v}$, where $\mathbf{a}$ is referred to as the primal and $\mathbf{v}$ the tangent. $\partial F_i$ form a linearized computation graph (third row in Fig. 3), that computes the JVP of the composition $\frac{\partial F}{\partial \mathbf{x}}\mathbf{v}$:

$$\frac{\partial F}{\partial \mathbf{x}}\mathbf{v} = \partial F(\mathbf{x})(\mathbf{v}) = [\partial F_L \circ \partial F_{L-1} \circ \cdots \circ \partial F_1](\mathbf{x})(\mathbf{v}). \tag{2}$$

By setting the tangent to $\mathbf{v}$ one of the standard basis of $\mathbb{R}^d$, JVP computes one column of the Jacobian $D_F$, so the full Jacobian can be computed with $d$ JVPs. Each JVP call requires $\mathcal{O}(\max(d, h))$ memory as only the current activation $\mathbf{y}_i$ and tangent $\mathbf{v}_i$ are needed to carry out the computation, and the computation complexity is usually in the same order as the forward computation graph. In the case of MLP, both the forward and the linearized graph have a complexity of $\mathcal{O}(dh + (L-1)h^2)$.

**Backward mode AD**   Each primitives $F_i$ is linearized as the adjoint of the Fréchet derivative $\partial^\top F_i$ instead, which computes the vector-Jacobian-product (VJP): $\partial^\top F_i(\mathbf{a})(\mathbf{v}^\top) = \mathbf{v}^\top \frac{\partial F}{\partial \mathbf{x}}\big|_{\mathbf{a}}$ where $\mathbf{v}^\top$ is the cotangent. The linearized computation graph now runs in the reverse order:

$$\mathbf{v}^\top \frac{\partial F}{\partial \mathbf{x}} = \partial^\top F(\mathbf{x})(\mathbf{v}^\top) = [\partial^\top F_1(\mathbf{x}) \circ \cdots \circ \partial^\top F_{L-1}(\mathbf{y}_{L-2}) \circ \partial^\top F_L(\mathbf{y}_{L-1})](\mathbf{v}^\top), \tag{3}$$

which is also clear from Fig. 3. Furthermore, due to this reversion, we first need to do a forward pass to obtain the evaluation trace $\{\mathbf{y}_i\}_{i=1}^L$ before we can invoke the VJPs $\partial^\top F_i$, which apparent as shown in Eq. 3. Hence the number of sequential computations is twice as much compared to forward mode. The memory requirement becomes $\mathcal{O}(d + (L-1)h)$ as we need to store the entire evaluation trace. Similar to JVP, VJP computes one row of $J_F$ at a time, so the full Jacobian $\frac{\partial F}{\partial \mathbf{x}}$ can be computed using $d'$ VJPs. When optimizing scalar cost functions $\ell(\theta) : \mathbb{R}^n \to \mathbb{R}$ of the network parameters $\theta$, backward mode efficiently trades off memory with computation complexity as $d' = 1$ and only 1 VJP is needed to get the full Jacobian. Furthermore, all parameter $\theta_i$ can use the same cotangent $\mathbf{v}^\top$, whereas with forward mode, separate tangent for each parameter $\theta_i$ is needed.

## 3.2   Inefficiency of the first-order AD for high-order derivative on inputs

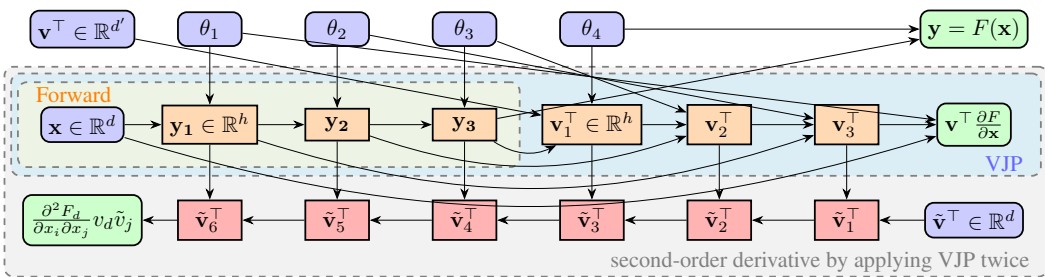

Figure 1: The computation graph of computing second order gradient by repeated application of backward mode AD, for a function $F(\cdot)$ with 4 primitives ($L = 4$), which computes the Hessian-vector-product. Red nodes represent the cotangent nodes in the second backward pass. With each repeated application of VJP the length of sequential computation doubles.

High-order input derivatives $\frac{\partial^k u_\theta}{\partial \mathbf{x}^k}$ for scalar $u_\theta$ can be implemented as repeated applications of first-order AD, but this approach will exhibit fundamental inefficiency that cannot be remedied by randomization.

**Repeating backward mode AD**   With each repeated application of backward mode AD, the new evaluation trace will include the cotangents from the previous application of backward AD, so the length of sequential computation **doubles**. Furthermore, the size of the cotangent also grows by $d$ times. Therefore applying backward mode AD has additional memory cost of $\mathcal{O}(d + (L-1)h)$ and

additional computation cost of $\mathcal{O}\big(2dh + 2(L-1)h^2\big)$, which is clear from Fig. 1. In general, with $k$ repeated applications of backward mode AD will incur $\mathcal{O}\big(2^{k-1}(d + (L-1)h)\big)$ memory cost and $\mathcal{O}\big(2^k(dh + (L-1)h^2)\big)$ computation cost. And $\mathcal{O}\big(d^{k-1}\big)$ calls are needed to evaluate the entire derivative tensor. So both memory and compute scale **exponentially** in derivative order $k$

**Repeating forward mode AD**   Consider $u_\theta : \mathbb{R}^d \to \mathbb{R}$. The input tangent dimension is $d$ on the first application of forward mode AD, but on the second application, it will become $d \times d$ since we are now computing the forward mode AD for $\nabla u_\theta : \mathbb{R}^d \to \mathbb{R}^d$. So the size of the input tangent with $k$ repeated application is $\mathcal{O}\big(d^k\big)$, so it grows **exponentially**. This is also inefficient.

**Mixed mode AD schemes are also likely inefficient**   See more detail in Appendix C.

### 3.3   Stochastic Dimension Gradient Descent

SDGD [13] amortizes high-dimensional differential operators by computing only a minibatch of derivatives in each iteration. It replaces a differential operator $\mathcal{D}$ with a randomly sampled subset of additive terms, where each term only depends on a few input dimensions

$$\mathcal{D} := \sum_{j=1}^{N_\mathcal{D}} \mathcal{D}_j \approx \frac{N_\mathcal{D}}{|J|} \sum_{j \in J} \mathcal{D}_j := \tilde{\mathcal{D}}_J, \tag{4}$$

where $\tilde{\mathcal{D}}_J$ denotes the SDGD operator that approximates the true operator $\mathcal{D}$, $J$ is the sampled index set, and $|J|$ is the batch size. For example, in $d$-dimensional Poisson equation, $N_\mathcal{D} = d$, $\mathcal{D} = \sum_{j=1}^{d} \frac{\partial^2}{\partial x_j^2}$, and the additive terms are $\mathcal{D}_j = \frac{\partial^2}{\partial x_j^2}$.

$\tilde{\mathcal{D}}_J$ are cheaper to compute than $\mathcal{D}$ due to reduced dimensionality: for each sampled index, by treating all other input as constant we get a function with scalar input and output. For a given index set $J$, the memory requirements are reduced from $\mathcal{O}\big(2^{k-1}(d + (L-1)h)\big)$ to $\mathcal{O}\big(|J|(2^{k-1}(1 + (L-1)h))\big)$, and the computation complexity reduces to $\mathcal{O}\big(|J|2^k(h + (L-1)h^2)\big)$. This reduction is significant when $d \gg h$ as in the experimental setting of SDGD [13], but the exponential scaling in $k$ persists.

### 3.4   Univariate Taylor mode AD

One way to define high-order AD is by determining how the high-order Taylor expansion of a univariate function changes when mapped by primitives. Firstly, the Fréchet derivative $\partial F$ can be rewritten to operate on a space curve $g : \mathbb{R} \to \mathbb{R}^d$ that passes through the primal $\mathbf{a}$, i.e. $g(t) = \mathbf{a}$, and has tangent $g'(t) = \mathbf{v}$:

$$\partial F(g(t))(g'(t)) = \left.\frac{\partial F}{\partial \mathbf{x}}\right|_{\mathbf{x}=g(t)} g'(t) = \frac{\mathrm{d}}{\mathrm{d}t}[F \circ g](t). \tag{5}$$

This shows that the $\partial$ (JVP) is the same as the univariate chain rule. The tuple $J_g(t) := (g(t), g'(t))$ can be thought of as the first-order expansion of $g$ which lives in the tangent bundle of $F$. Treating $F$ as the smooth map between manifolds, we can define the pushforward $\mathrm{d}F$ which pushes the first order expansion of $g$ (i.e. $J_g(t)$) forward to the first order expansion of $F \circ g$ (i.e. $J_{F \circ g}(t)$):

$$\mathrm{d}F(J_g(t)) = J_{F \circ g}(t) = \left([F \circ g](t), \frac{\mathrm{d}}{\mathrm{d}t}[F \circ g](t)\right) = (F(\mathbf{a}), \partial F(\mathbf{a})(\mathbf{v})). \tag{6}$$

Naturally, to extend this to higher orders, one can consider the $k$th order expansion of the input curve $g$, which is equivalent to the tuple $J_g^k(t) := (g(t), g'(t), g''(t), \ldots, g^{(k)}(t)) = (\mathbf{a}, \mathbf{v}^{(1)}, \mathbf{v}^{(2)}, \ldots, \mathbf{v}^k)$ known as the $k$-jet of $g$ where $\mathbf{v}^j$ is called the $j$th order tangent of $g$. $J_g^k$ lives in the $k$th order tangent bundle of $F$, and we can define the $k$th-order pushforward $\mathrm{d}^k F$ :

$$\mathrm{d}^k F(J_g^k(t)) = J_{F \circ g}^k(t) = \left([F \circ g](t), \frac{\partial}{\partial t}[F \circ g](t), \frac{\partial^2}{\partial t^2}[F \circ g](t), \ldots, \frac{\partial^k}{\partial t^k}[F \circ g](t)\right)$$
$$= (F(\mathbf{a}), \partial F(\mathbf{a})(\mathbf{v}^{(1)}), \partial^2 F(\mathbf{a})(\mathbf{v}^{(1)}, \mathbf{v}^{(2)}), \ldots, \partial^k F(\mathbf{a})(\mathbf{v}^{(1)}, \ldots, \mathbf{v}^{(k)})), \tag{7}$$

which pushes the $k$th order expansion of $g$ (i.e. $J_g^k$) forward to the $k$th order expansion of $F \circ g$ (i.e. $J_{F \circ g}^k$). $\partial^k F = \frac{\partial^k}{\partial t^k}[F \circ g](t)$ is the $k$-th order Fréchet derivative, whose analytical formula is given by the high-order univariate chain rule known as the Faa di Bruno's formula (Eq. 43).

Since $J_g^k$ contains all information needed to evaluate $\frac{\partial^j}{\partial t^j}[F \circ g](t)$ for any $j \leq k$, the map $\mathrm{d}^k F$ is well-defined. $\mathrm{d}^k$ defines a high-order AD: we can compute $\mathrm{d}^k F$ of arbitrary composition $F$ from the $k$th-order pushforward of the primitives $\mathrm{d}^k F_i$, since $\mathrm{d}^k$ is an homomorphism of the group $(\{F_i\}, \circ)$:

$$\mathrm{d}^k[F_2 \circ F_1](J_g^k(t)) = J_{F_2 \circ F_1 \circ g}^k(t) = \mathrm{d}^k F_2(J_{F_1 \circ g}^k(t)) = [\mathrm{d}^k F_2 \circ \mathrm{d}^k F_1](J_g^k(t)). \tag{8}$$

This approach of composing $\mathrm{d}^k$ of primitives is also known as the Taylor mode AD. For more details on Taylor mode AD, see Appendix D.

## 4    Method

From the previous discussion, it is clear that the exponential scaling in $k$ for the problem described in Eq. 1 cannot be mitigated by amortization alone. Although high-order AD methods like Taylor mode AD [6] can address this scaling issue, it is only defined for univariate functions. In this section, we describe a method that addresses the scaling issue in $k$ and $d$ simultaneously when amortizing Eq. 1 by seeing univariate Taylor mode AD as contractions of multivariate derivative tensor.

### 4.1    Univariate Taylor mode AD as contractions of multivariate derivative tensor

$\mathrm{d}F$ projects the Jacobian of $F$ to $\mathbb{R}^{d'}$ with a 1-jet $J_g(t)$. Similarly, $\mathrm{d}^k F$ contracts a set of derivative tensors to $\mathbb{R}^{d'}$ with a $k$-jet $J_g^k$. We can expand $\frac{\partial^k}{\partial t^k} F \circ g$ with Eq. 43 to see the form of the contractions. For example, $\partial F$ is JVP, and $\partial^2 F$ contains a quadratic form of the Hessian $D_F^2$:

$$\partial^2 F(\mathbf{a})(\mathbf{v}^{(1)}, \mathbf{v}^{(2)}) = \frac{\partial^2}{\partial t^2}[F \circ g](t) = D_F(\mathbf{a})\mathbf{v}^{(2)} + D_F^2(\mathbf{a})_{d', d_1, d_2} v_{d_1}^{(1)} v_{d_2}^{(1)}. \tag{9}$$

From Eq. 43, one can always find a $J_g^l$ with large enough $l \geq k$ such that there exists $k \leq l' \leq l$ with $\partial^{l'} F(J_g^{l'}) = D_F^k(\mathbf{a}) \cdot \otimes_{i=1}^k \mathbf{v}^{(v_i)}$ where $v_i \in [1, k]$, by setting some tangents $\mathbf{v}^{(v_i)}$ to the zero vector. That is, arbitrary derivative tensor contraction is contained within a Fréchet derivative of high-order, which can be efficiently evaluated through Taylor mode AD.

How large $l$ should be depends on how off-diagonal the operator is. If the operator is diagonal (i.e. contains no mixed partial derivatives), $l = k$ is enough. If the operator is maximally non-diagonal, i.e. it is a partial derivative where all dimensions to be differentiated are distinct, then the minimum $l$ needed is $(1 + k)k/2$. For more details, please refer to Appendix F where a general procedure for determining the jet structure is discussed.

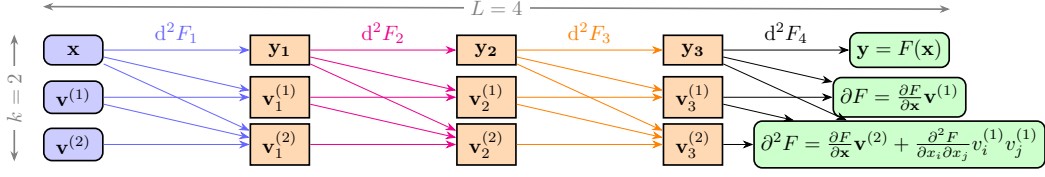

Figure 2: The computation graph of $\mathrm{d}^2 F$ for $F$ with 4 primitives. Parameters $\theta_i$ are omitted. The first column from the left represents the input 2-jet $J_g^2(t) = (\mathbf{x}, \mathbf{v}^{(1)}, \mathbf{v}^{(2)})$, and $\mathrm{d}^2 F_1$ pushes it forward to the 2-jet $J_{F_1 \circ g}^2(t) = (\mathbf{y}_1, \mathbf{v}_1^{(1)}, \mathbf{v}_1^{(2)})$ which is the subsequent column. Each row can be computed in parallel, and no evaluate trace needs to be cached.

### 4.2    Estimating arbitrary differential operator by pushing forward random jets

Next, we show how to use the above facts to construct a stochastic estimator derivative operator. Differential operators can be evaluated through derivative tensor contraction. The action of the derivative $\mathcal{D}^\alpha = \frac{\partial^{|\alpha|}}{\partial x_1^{\alpha_1}, \dots, \partial x_d^{\alpha_d}}$ on function $u$ can be identified with the derivative tensor slice $D_u^{|\alpha|}(\mathbf{a})_\alpha$. Differential operator $\mathcal{L}$ can be written as a linear combination of derivatives: $\mathcal{L} = \sum_{\alpha \in \mathcal{I}(\mathcal{L})} C_\alpha \mathcal{D}^\alpha$, where $\mathcal{I}(\mathcal{L})$ is the set of tensor indices representing terms included in the operator $\mathcal{L}$. For simplicity we only consider $k$th order differential operator, i.e. $|\alpha| = k \in \mathbb{N}$ for all $\alpha$. For scalar $u : \mathbb{R}^d \to \mathbb{R}$,

we can identify a $k$th order differential operator $\mathcal{L}$ with the following tensor dot product

$$\mathcal{L}u(\mathbf{a}) = \sum_{\alpha \in \mathcal{I}(\mathcal{L})} C_\alpha \mathcal{D}^\alpha u(\mathbf{a}) = \sum_{d_1,\ldots,d_k} D_u^k(\mathbf{a})_{d_1,\ldots,d_k} C_{d_1,\ldots,dk}(\mathcal{L}) = D_u^k(\mathbf{a}) \cdot \mathbf{C}(\mathcal{L}),$$
(10)

where $d_i \in [1, d], i \in [1, k]$ is the tensor index on the $i$th axis, , and $\mathbf{C}(\mathcal{L})$ is a tensor of the same shape as $D_u^k(\mathbf{a})$ that equals $C_\alpha$ when $d_1, \ldots, d_k$ matches the multi-index $\alpha \in \mathcal{I}(\mathcal{L})$ and 0 otherwise. We call $\mathbf{C}(\mathcal{L})$ the coefficient tensor of $\mathcal{L}$. For example, the coefficient tensor of the Laplacian $\nabla^2$ is the $d$-dimensional identity matrix $\mathbf{I}$. More complicated operators can be built as $f(\mathbf{x}, u, \mathcal{D}_{k_1}u, \ldots, \mathcal{D}_{k_n}u)$ where $f$ is arbitrary function.

*Any derivative tensor contractions $D_u^k(\mathbf{a}) \cdot \mathbf{C}(\mathcal{L})$ can be estimated through random contraction, which can be implemented efficiently as pushing forward random jets from an appropriate distribution.* With random $(\mathbf{v}^{(1)}, \ldots, \mathbf{v}^{(k)})$, we have

$$\mathbb{E}[D_u^k(\mathbf{a})_{d_1,\ldots,d_k} v_{d_1}^{(v_1)} \ldots v_{d_k}^{(v_k)}] = D_u^k(\mathbf{a})_{d_1,\ldots,d_k} \mathbb{E}[v_{d_1}^{(v_1)} \ldots v_{d_k}^{(v_k)}] = D_u^k(\mathbf{a}) \cdot \mathbb{E}\left[\otimes_{i=1}^k \mathbf{v}^{(v_i)}\right] \quad (11)$$

where $\otimes$ denotes Kronecker product, $v_{d_i}^{(v_i)} \in [1, k]$ is the $d_i$ dimension of the $v_i$th order tangent in the input $k$-jet. Eq. 11 is an unbiased estimator of the $k$th order operator $\mathcal{L}u = D_u^k(\mathbf{a}) \cdot \mathbf{C}(\mathcal{L})$ when

$$\mathbb{E}[v_{d_1}^{(v_1)} \ldots v_{d_k}^{(v_k)}] = C_{d_1,\ldots,dk}(\mathcal{L}).$$
(12)

For example, the condition for unbiasedness for the Laplacian $\nabla^2$ is $\mathbb{E}[\mathbf{v}^{(a)}\mathbf{v}^{(b)\top}] = \mathbf{I}$. As discussed, one can always find a $J_g^l$ with large enough $l \geq k$ such that $\partial^l F(J_g^l) = D_F^k(\mathbf{a}) \cdot \otimes_{i=1}^k \mathbf{v}^{(v_i)}$, so with a distribution $p$ over the input $l$-jet $J_g^l$ that satisfies the unbiasedness condition (Eq. 12), we have

$$\mathbb{E}_{J_g^l \sim p}[\partial^l u(J_g^l)] = \mathbb{E}[v_{d_1}^{(v_1)} \ldots v_{d_k}^{(v_k)}] = D_u^k(\mathbf{a}) \cdot \mathbf{C}(\mathcal{L}) = \mathcal{L}u(\mathbf{a}),$$
(13)

which means $\mathcal{L}u(\mathbf{a})$ can be approximated by the sample mean of the pushforwards of random $l$-jet drawn from $p$, which can be computed efficiently via Taylor mode AD. We call this method *Stochastic Taylor Derivative Estimator (STDE)*. The **advantages** of STDE are:

1. General: STDE can be applied to differential operators of arbitrary order and dimensionality.

2. Scalable: The scaling issue in the dimensionality $d$ and the derivative order $k$ are addressed at the same time. From the example computation graph (Fig. 2) we see that, for one call to $\mathrm{d}^k F$, the memory requirement has scaling of $\mathcal{O}(kd)$ and the computation complexity has scaling $\mathcal{O}(k^2 dL)$. Like first-order forward mode AD, the derivative tensor $D_u^k$ is never fully computed and stored. Combined with randomization, the polynomial scaling in $d$ will be removed.

3. Parallelizable: The number of sequential computations does not grow with the order as can be seen in Fig. 2, and the computation can be trivially vectorized and parallelized since the pushforward of sample jets can be computed independently, and it uses the same computation graph ($\mathrm{d}^k u$);

### 4.3 Constructing STDE for high-order differential operators with sparse random jets

Note that all coefficient tensor has the following additive decomposition:

$$\mathbf{C}(\mathcal{L}) = \sum_{d_1,\ldots,d_k \in \mathcal{I}(\mathcal{D})} C_{d_1,\ldots,d_k} \mathbf{e}_{d_1} \otimes \cdots \otimes \mathbf{e}_{d_k}$$
(14)

where $\mathbf{e}_i$ is the $i$th standard basis. For example, if the input dimension $d$ is 3, then $\mathbf{e}_2 = [0, 1, 0]^\top$. As discussed before, there exists a $J_g^k$ whose pushforward under $\partial^l u$ is equivalent to contracting $D_u^k$ with $\otimes_{i=1}^k \mathbf{e}_{d_i}$. We call $k$-jet consisting of only standard basis and the zero vector $\mathbf{0}$ *sparse*. Therefore the discrete distribution $p$ over the *sparse* $k$-jets in Eq. 14 satisfies the unbiasedness condition 12

$$p(\otimes_{i=1}^k \mathbf{e}_{d_i}) = C_{d_1,\ldots,d_k}/Z, \quad d_1, \ldots, d_k \in \mathcal{I}(\mathcal{L}),$$
(15)

where $Z$ is the normalization factor and we identify $\otimes_{i=1}^k \mathbf{e}_{d_i}$ with the corresponding $k$-jet $J_u^k$.

#### 4.3.1 Differential operator with easy to remove mixed partial derivatives

Next, we show some concrete examples for constructing STDE with sparse random jets.

**Laplacian**  From Eq. 9 we know that the quadratic form of Hessian can be computed through $\partial^2$ by setting $\mathbf{v}^{(2)} = \mathbf{0}$ and $\mathbf{v}^{(1)} = \mathbf{e}_j$. Therefore, the STDE of the Laplacian operator is given by

$$\tilde{\nabla}^2{}_J u_\theta(\mathbf{a}) = \frac{d}{|J|} \sum_{j \in J} \frac{\partial^2}{\partial x_j^2} u_\theta(\mathbf{a}) = \frac{d}{|J|} \sum_{j \in J} \partial^2 u_\theta(\mathbf{a})(\mathbf{e}_j, \mathbf{0}) = \frac{d}{|J|} \sum_{j \in J} \mathrm{d}^2 u_\theta(\mathbf{a}, \mathbf{e}_j, \mathbf{0})_{[2]} \quad (16)$$

where $J$ is the sampled index set, and the subscript $[2]$ means taking the second-order tangent from the output jet. See example implementation in JAX in Appendix A.4.

**High-order *diagonal* differential operators**  We call a differential operator *diagonal* if it is a linear combination of diagonal elements from the derivative tensor: $\mathcal{L} = \sum_{j=1}^{d} \frac{\partial^k}{\partial x_j^k}$. From Eq. 43 we see that setting the first-order tangent $\mathbf{v}^{(1)}$ to $\mathbf{e}_j$ and all other tangents $\mathbf{v}^{(i)}$ to the zero vector gives the desired high-order diagonal element:

$$\tilde{\mathcal{L}}_J u_\theta(\mathbf{a}) = \frac{d}{|J|} \sum_{j \in J} \frac{\partial^k}{\partial \mathbf{x}_j^k} u_\theta(\mathbf{a}) = \frac{d}{|J|} \sum_{j \in J} \partial^k u_\theta(\mathbf{a})(\mathbf{e}_j, \mathbf{0}, \dots). \quad (17)$$

**Second-order parabolic PDEs**  Second-order parabolic PDEs are a large class of PDEs. It includes the Fokker-Planck equation in statistical mechanics to describe the evolution of the state variables in stochastic differential equations (SDEs), which can be used for generative modeling [38]. It also includes the Black-Scholes equation in mathematical finance for option pricing, the Hamilton-Jacobi-Bellman equation in optimal control, and the Schrödinger equation in quantum physics and chemistry. Its form is given by

$$\frac{\partial}{\partial t} u(\mathbf{x}, t) + \frac{1}{2} \operatorname{tr}\left( \sigma \sigma^\top (\mathbf{x}, t) \frac{\partial^2}{\partial \mathbf{x}^2} u(\mathbf{x}, t) \right) + \nabla u(\mathbf{x}, t) \cdot \mu(\mathbf{x}, t) + f(t, \mathbf{x}, u(\mathbf{x}, t), \sigma^\top (\mathbf{x}, t) \nabla u(\mathbf{x}, t)) = 0. \quad (18)$$

We have a second order derivative term $\frac{1}{2} \operatorname{tr}\left( \sigma(\mathbf{x}, t) \sigma(\mathbf{x}, t)^\top \frac{\partial^2}{\partial \mathbf{x}^2} u(\mathbf{x}, t) \right)$ with *off-diagonal* term. The off-diagonals can be easily removed via a change of variable:

$$\frac{1}{2} \operatorname{tr}\left( \sigma(\mathbf{x}, t) \sigma(\mathbf{x}, t)^\top \frac{\partial^2}{\partial \mathbf{x}^2} u(\mathbf{x}, t) \right) = \frac{1}{2} \sum_{i=1}^{d} \partial^2 u(\mathbf{x}, t)(\sigma(\mathbf{x}, t) \mathbf{e}_i, \mathbf{0}). \quad (19)$$

See derivation in Appendix E. Its STDE samples over the $d$ terms in the expression above.

### 4.3.2  Differential operators with arbitrary mixed partial derivative

It is not always possible to remove the mixed partial derivatives but discussed in section 4.2, for an arbitrary $k$th order derivative tensor element $D_u^k(\mathbf{a})_{n_1, \dots, n_k}$, we can find an appropriate $l$-jet $J_g^l(t)$ with $g(t) = \mathbf{a}$ such that $\partial^l u(J_g^l) = D_u^k(\mathbf{a})_{n_1, \dots, n_k}$. Here we show a concrete example.

**2D Korteweg-de Vries (KdV) equation**  Consider the following 2D KdV equation

$$u_{ty} + u_{xxxy} + 3(u_y u_x)_x - u_{xx} + 2u_{yy} = 0. \quad (20)$$

All the derivative terms can be found in the pushforward of the following jet:

$$\mathfrak{J} = \mathrm{d}^{13} u(\mathbf{x}, \mathbf{v}^{(1)}, \dots, \mathbf{v}^{(13)}), \ \mathbf{v}^{(3)} = \mathbf{e}_x, \mathbf{v}^{(4)} = \mathbf{e}_y, \mathbf{v}^{(7)} = \mathbf{e}_t, \mathbf{v}^{(i)} = \mathbf{0}, \forall i \notin \{3, 4, 7\},$$
$$u_x = \mathfrak{J}_{[1]}, \ u_y = \mathfrak{J}_{[2]}, \ u_{xx} = \mathfrak{J}_{[4]}, \ u_{xy} = \mathfrak{J}_{[5]}/35, \quad (21)$$
$$u_{yy} = \mathfrak{J}_{[6]}/35, \ u_{ty} = \mathfrak{J}_{[9]}/330, \ u_{xxxy} = \mathfrak{J}_{[11]}/200200.$$

where the subscript $[i]$ means selecting the $i$th order tangent from the jet, and the prefactors are determined through Faa di Bruno's formula (Eq. 43). In this case, no randomization is needed since all the terms can be computed with just one pushforward. Alternatively, these terms can be computed with pushforwards of different jets of lower order (Appendix I.4). When input dimension $d$ is high, randomization via STDE will provide significant speed up. We tested a few more high-order PDEs with irremovable mixed partial derivatives (see Appendix I.4), and the experimental results will be provided later.

## 4.4 Dense random jet and connection to HTE

In section 4.3 we show how to construct STDE with the pushforward of *sparse* random jets. It is also possible to construct STDE with *dense* random jets, i.e. jets with tangents that are not the standard basis. For example, the classical method of Hutchinson trace estimator (HTE) [16] can be implemented in the STDE framework as the pushforward of isotropic dense random jets, i.e. $(\mathbf{a}, \mathbf{v}, \mathbf{0}) \sim \delta_\mathbf{a} \times p \times \delta$ with $\mathbb{E}_p[\mathbf{v}\mathbf{v}^\top] = \mathbf{I}$.

We generalize the dense construction to **arbitrary second-order differential operators** using a multivariate Gaussian distribution with the eigenvalues of the corresponding coefficient tensor as its covariance. Suppose $\mathcal{D}$ is a second-order differential operator with coefficient tensor $\mathbf{C}$. With the eigendecomposition $\mathbf{C}'' = \frac{1}{2}(\mathbf{C} + \mathbf{C}^\top) + \lambda\mathbf{I} = \mathbf{U}\boldsymbol{\Sigma}\mathbf{U}^\top$ where $-\lambda$ is smaller than the smallest eigenvalue of $\mathbf{C}$ , we can construct a STDE for $\mathcal{D}$:

$$\mathbb{E}_{\mathbf{v}\sim\mathcal{N}(\mathbf{0},\boldsymbol{\Sigma})}[\partial^2 u(\mathbf{a})(\mathbf{U}\mathbf{v}, \mathbf{0})] - \lambda\mathbb{E}_{\mathbf{v}\sim\mathcal{N}(\mathbf{0},\mathbf{I})}[\partial^2 u(\mathbf{a})(\mathbf{v}, \mathbf{0})] = D_u^2(\mathbf{a}) \cdot [\mathbf{C}'' - \lambda\mathbf{I}] = D_u^2(\mathbf{a}) \cdot \mathbf{C}. \quad (22)$$

However, it is not always possible to construct dense STDE beyond the second order, even if we consider $p$ with non-diagonal covariance. We prove this by providing a counterexample: one cannot construct an STDE for the fourth order operator $\sum_{i=1}^d \frac{\partial^4}{\partial x^4}$ with dense jets. For more details on dense jets, see Appendix K. For specific high-order operators like the Biharmonic operator, it is still possible to construct STDE with dense jets which we show in Appendix J.

The main differences between the sparse and the dense version of STDE are:

1. sparse STDE is universally application whereas the dense STDE can only be applied to certain operators;

2. the source of variance is different (see Appendix K.3).

It is also worth noting that both the sparse and the dense versions of STDE would have similar computation costs if the batch size of random jets were the same. In general, we would suggest to use sparse STDE unless it is known a priori that the sparse version would suffer from excess variance and the dense STDE is applicable.

## 5 Experiments

We applied STDE to amortize the training of PINNs on a set of real-world PDEs. For the case of $k = 2$ and large $d$, we tested two types of PDEs: inseparable and effectively high-dimensional PDEs (Appendix I.1) and semilinear parabolic PDEs (Appendix I.2). We also tested high-order PDEs (Appendix I.4) that cover the case of $k = 3, 4$, which includes PDEs describing 1D and 2D nonlinear dynamics, and high-dimensional PDE with gradient regularization [42]. Furthermore, we tested a weight-sharing technique (Appendix G), which further reduces memory requirements (Appendix I.3). In all our experiments, STDE drastically reduces computation and memory costs in training PINNs, compared to the baseline method of SDGD with stacked backward-mode AD. Due to the page limit, the most important results are reported here, and the full details including the experiment setup and hyperparameters (Appendix H) can be found in the Appendix.

### 5.1 Physics-informed neural networks

PINN [33] is a class of neural PDE solver where the ansatz $u_\theta(\mathbf{x})$ is parameterized by a neural network with parameter $\theta$. It is a prototypical case of the optimization problem in Eq. 1. We consider PDEs defined on a domain $\Omega \subset \mathbb{R}^d$ and boundary/initial $\partial\Omega$ as follows

$$\mathcal{L}u(\mathbf{x}) = f(\mathbf{x}), \quad \mathbf{x} \in \Omega, \quad \mathcal{B}u(\mathbf{x}) = g(\mathbf{x}), \quad \mathbf{x} \in \partial\Omega, \quad (23)$$

where $\mathcal{L}$ and $\mathcal{B}$ are known operators, $f(\mathbf{x})$ and $g(\mathbf{x})$ are known functions for the residual and boundary/initial conditions, and $u : \mathbb{R}^d \to \mathbb{R}$ is a scalar-valued function, which is the unknown solution to the PDE. The approximated solution $u_\theta(\mathbf{x}) \approx u(\mathbf{x})$ is obtained by minimizing the mean squared error (MSE) of the PDE residual $R(\mathbf{x}; \theta) = \mathcal{L}u_\theta(\mathbf{x}) - f(\mathbf{x})$:

$$\ell_{\text{residual}}(\theta; \{\mathbf{x}^{(i)}\}_{i=1}^{N_r}) = \frac{1}{N_r} \sum_{i=1}^{N_r} \left| \mathcal{L}u_\theta(\mathbf{x}^{(i)}) - f(\mathbf{x}^{(i)}) \right|^2 \quad (24)$$

where the residual points $\{\mathbf{x}^{(i)}\}_{i=1}^{N_r}$ are sampled from the domain $\Omega$. We use the technique from [26] that reparameterizes $u_\theta$ such that the boundary/initial condition $\mathcal{B}u(\mathbf{x}) = g(\mathbf{x})$ are satisfied exactly for all $\mathbf{x} \in \partial\Omega$, so boundary loss is not needed.

**Amortized PINNs**   PINN training can be amortized by replacing the differential part of the operator $\mathcal{L}$ with a stochastic estimator like SDGD and STDE. For example, for the Allen-Cahn equation, $\mathcal{L}u = \nabla^2 u + u - u^3$, the differential part of $\mathcal{L}$ is the Laplacian $\nabla^2$. With amortization, we minimize the following loss

$$\tilde{\ell}_{\text{residual}}(\theta; \{\mathbf{x}^{(i)}\}_{i=1}^{N_r}, J, K) = \frac{1}{N_r} \sum_{i=1}^{N_r} \left[ \tilde{\mathcal{L}}_J u_\theta(\mathbf{x}^{(i)}) - f(\mathbf{x}^{(i)}) \right] \cdot \left[ \tilde{\mathcal{L}}_K u_\theta(\mathbf{x}^{(i)}) - f(\mathbf{x}^{(i)}) \right], \quad (25)$$

which is a modification of Eq. 24. Its gradient $\frac{\partial \tilde{\ell}_{\text{residual}}}{\partial \theta}$ is then an unbiased estimator to the gradient of the original PINN residual loss, i.e. $\mathbb{E}[\frac{\partial \tilde{\ell}_{\text{residual}}}{\partial \theta}] = \frac{\partial \ell_{\text{residual}}}{\partial \theta}$.

## 5.2   Ablation study on the performance gain

To ascertain the source performance gain of our method, we conduct a detailed ablation study on the inseparable Allen-Cahn equation with a two-body exact solution described in Appendix I.1. The results are in Table 1 and 2, where the best results for each dimensionality are marked in bold. All methods were implemented using JAX unless stated. OOM indicates that the memory requirement exceeds 40GBs. Since the only change is how the derivatives are computed, the relative L2 error is expected to be of the same order among different randomization methods, as seen in Table 3 in the Appendix. We have included Forward Laplacian which is an exact method. It is expected to perform better in terms of L2 error. However, as we can see in Table 3, the L2 error is of the same order, at least in the case where the dimension is more than 1000.

Table 1: Speed ablation for the two-body Allen-Cahn equation.

| Speed (it/s) ↑ | 100 D | 1K D | 10K D | 100K D | 1M D |
|---|---|---|---|---|---|
| Backward mode SDGD (PyTorch) [13] | 55.56 | 3.70 | 1.85 | 0.23 | OOM |
| Backward mode SDGD | 40.63 | 37.04 | 29.85 | OOM | OOM |
| Parallelized backward mode SDGD | 1376.84 | 845.21 | 216.83 | 29.24 | OOM |
| Forward-over-Backward SDGD | 778.18 | 560.91 | 193.91 | 27.18 | OOM |
| Forward Laplacian [24] | **1974.50** | 373.73 | 32.15 | OOM | OOM |
| **STDE** | 1035.09 | **1054.39** | **454.16** | **156.90** | **13.61** |

Table 2: Memory ablation for the two-body Allen-Cahn equation.

| Memory (MB) ↓ | 100 D | 1K D | 10K D | 100K D | 1M D |
|---|---|---|---|---|---|
| Backward mode SDGD (PyTorch) [13] | 1328 | 1788 | 4527 | 32777 | OOM |
| Backward mode SDGD | 553 | 565 | 1217 | OOM | OOM |
| Parallelized backward mode SDGD | 539 | 579 | 1177 | 4931 | OOM |
| Forward-over-Backward SDGD | 537 | 579 | 1519 | 4929 | OOM |
| Forward Laplacian [24] | **507** | 913 | 5505 | OOM | OOM |
| **STDE** | 543 | **537** | **795** | **1073** | **6235** |

**JAX vs PyTorch**   The original SDGD with stacked backward mode AD was implemented in PyTorch. We reimplement it in JAX (see Appendix A.1). From Table 1 and 2, JAX provides $\sim 15\times$ speed-up and up to $\sim 4\times$ memory reduction.

**Parallelization**   The original SDGD implementation uses a for-loop to iterate through the sampled dimension (Appendix A.1). This can be parallelized (denoted as "Parallelized SDGD via HVP", details in Appendix A.2). Parallelization provides $\sim 15\times$ speed up and reduction in peak memory for the JIT compilation phase. We also tested mixed mode AD (dubbed as "Forward-over-Backward SDGD"), which gives roughly the same performance as parallelized stacked backward mode, which is expected as explained in Appendix C.

**Forward Laplacian**   Forward Laplacian [24] provides a constant-level optimization for the calculation of Laplacian operator by removing the redundancy in the AD pipeline, and we can see from

Table 1 and 2 that it is the best method in both speed and memory when the dimension is 100. But since it is not a randomized method, the scaling is much worse. Its computation complexity is $\mathcal{O}(d)$, whereas a randomized estimator like STDE has a computation complexity of $\mathcal{O}(|J|)$. Naturally, with a high enough input dimension $d$, the difference in the constant prefactor is trumped by scaling. When the dimension is larger than 1000, it becomes worse than even parallelized stacked backward mode SDGD.

**STDE**    Compared to the best realization of baseline method SDGD, the parallelized stacked backward mode AD, STDE provides up to $10\times$ speed up and memory reduction of at least $4\times$.

## 6    Conclusion

We introduce STDE, a general method for constructing stochastic estimators for arbitrary differential operators that can be evaluated efficiently via Taylor mode AD. We evaluated STDE on PINNs, an instance of the optimization problem where the loss contains differential operators. Amortization with STDE outperforms the baseline methods, and STDE also applies to a wider class of problems as it can be applied to arbitrary differential operators.

**Applicability**    Besides PINNs, STDE can be applied to arbitrarily high-order and high-dimensional AD-based PDE solvers. This makes STDE more general than a branch of related methods. STDE is also more applicable than deep ritz method [41], weak adversarial network (WAN) [43], backward SDE-based solvers [3, 34, 10], deep Galerkin method [35], and the recently proposed forward Laplacian [24], which are all restricted to specific forms of second-order PDEs. STDE applies naturally to differential operators in PDEs, but it can also be applied to other problems that require input gradients. For example, adversarial attacks, feature attribution, and meta-learning, to name a few.

**Limitations**    Being a general method, STDE forgoes the optimization possibilities that apply to specific operators. Furthermore, we did not consider variance reduction techniques that could be applied, which can be explored in future works. Also, we observed that lowering the randomization batch size improves both speed and memory profile, but the trade-off between cheaper computation and larger variance needs further analysis. Furthermore, the method is not suited for computing the high order derivative of neural network parameter as explained in Section 3.

**Future works**    The key insight of the STDE construction is that the univariate Taylor mode AD contains arbitrary contraction of the derivative tensor and that derivative operators are derivative tensor contractions. This shows the connection between the fields of AD and randomized numerical linear algebra and indicates that further works in the intersection of these two fields might bring significant progress in large-scale scientific modeling with neural networks. One example would be the many-body Schrödinger equations, where one needs to compute a high-dimensional Laplacian. Another example is the high-dimensional Black-Scholes equation, which has numerous uses in mathematical finance.

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

# A  Example implementations

## A.1  PyTorch implementation of SDGD-PINN using backward mode AD

The original implementation of SDGD-PINN [13] computes the SDGD estimation of derivatives using a for-loop that iterates over the sampled PDE term/dimension. For example, given a function $f$ representing the MLP PINN, the computation of SDGD for the Laplacian operator can be implemented in PyTorch as follows:

```
f_x = torch.autograd.grad(f.sum(), x, create_graph=True)[0]
idx_set = np.random.choice(dim, sdgd_batch_size, replace=False)
hess_diag_val = 0.
for i in idx_set:
    hess_diag_i = torch.autograd.grad(
        f_x[:, i].sum(), x, create_graph=True)[0][:, i]
    hess_diag_val += hess_diag_i.detach() * dim / sdgd_batch_size
```

After computing the PDE differential operator, it is plugged into the residual loss, and then backward-mode AD is employed to produce the gradient for optimization concerning $\theta$.

## A.2  JAX implementation of SDGD Parallelization via HVP

```
def hvp(f, x, v):
  """stacked backward-mode Hessian-vector product"""
  return jax.grad(lambda x: jnp.vdot(jax.grad(f)(x), v))(x)

f_hess_diag_fn = lambda i: hvp(f_partial, x_i, jnp.eye(dim)[i])[i]
idx_set = jax.random.choice(
  key, dim, shape=(sdgd_batch_size,), replace=False
)
hess_diag_val = jax.vmap(f_hess_diag_fn)(idx_set)
```

## A.3  JAX implementation of Forward-over-backward AD

The forward-over-backward AD In JAX mentioned in Appendix C can be implemented as follows:

```
f_grad_fn = jax.grad(f)
f_x, f_hess_fn = jax.linearize(f_grad_fn, x_i)   # jvp over vjp
f_hess_diag_fn = lambda i: f_hess_fn(jnp.eye(dim)[i])[i]
hess_diag_val = jax.vmap(f_hess_diag_fn)(idx_set)
```

## A.4  JAX implementation of STDE for the Laplacian operator

```
idx_set = jax.random.choice(
  key, dim, shape=(batch_size,), replace=False
)
rand_jet = jax.vmap(lambda i: jnp.eye(dim)[i])(idx_set)
pushfwd_2_fn = lambda v: jet.jet(
  fun=fn, primals=(x,), series=((v, jnp.zeros(dim)),)
) # pushforward of the 2-jet (x, v, 0), i.e. \dd^2 f(x, v, 0)
f_vals, (_, vhv) = jax.vmap(pushfwd_2_fn)(rand_jet)
hess_diag_val = dim / batch_size * vhv
```

The `jet.jet` function from JAX implements the high-order pushforward $\mathrm{d}^n$ of jets in Eq. 7. It decomposes the input function into primitives, which have analytical derivatives derived up to arbitrary order, and uses the generalized chain rule (see section D.2) to compose the primitives into the pushforward of jets. Note that in the API of `jet.jet`, all the high-order tangents of the input jet are specified via the `series` argument.

# B Further details on first-order auto-differentiation

## B.1 Computation graph of first-order AD

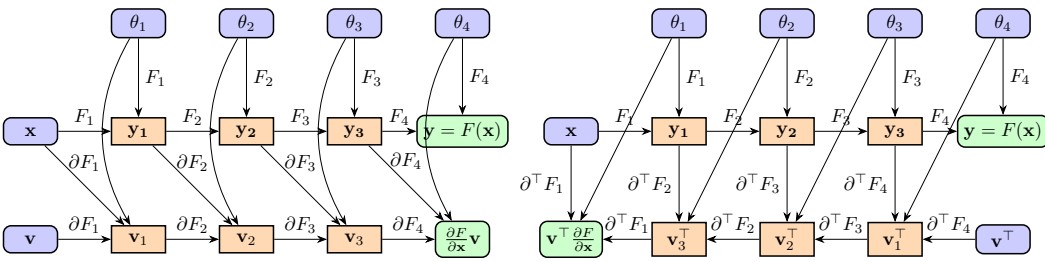

Figure 3: The computation graph of forward mode AD (left) and backward mode AD (right) of a function $F(\cdot)$ with 4 primitives $F_i$ each parameterized by $\theta_i$. Nodes represent (intermediate) values, and arrows represent computation. Input nodes are colored blue; output nodes are colored green, and intermediate nodes are colored yellow.

## B.2 Derivative via composition

First-order AD is based on a simple observation: for a set of functions $\mathcal{L}$, the set of tuples of functions $f$ and its *Jacobian* $J_f$ is *closed under composition*:

$$(f, J_f) \circ (g, J_g) = (f \circ g, J_{f \circ g}), \quad J_{f \circ g}(t) = J_f(g(t))J_g(t) \tag{26}$$

where $\circ$ denotes both function composition and the composition of the tuple $(f, J_f)$. If we have the analytical formula of the Jacobian $J_f$ for every $f \in \mathcal{L}$, then we can calculate the Jacobian of any composition of functions from $\mathcal{L}$ using the above composition rule for the tuple $(f, J_f)$. The set $\mathcal{L}$ of functions are usually called the *primitives*.

## B.3 Fréchet derivative and linearization

Given normed vector spaces $V, W$, the Fréchet derivative $\partial f$ of a function $f : V \to W$ is a map from $V$ to the space of all bounded linear operators from $V$ to $W$, denoted as $\mathrm{L}(V, W)$, that is

$$\partial f : V \to \mathrm{L}(V, W), \tag{27}$$

such that at a point $\mathbf{a} \in V$ it gives the *best linear approximation* $\partial f(\mathbf{a})(\cdot)$ of $f$, in the sense that

$$\lim_{\|\mathbf{h}\| \to 0} \frac{\|f(\mathbf{a} + \mathbf{h}) - f(\mathbf{a}) - \partial f(\mathbf{a})(\mathbf{h})\|_W}{\|h\|_V} = 0 \tag{28}$$

Therefore, it is also called the *linearization* of $f$ at point $\mathbf{a}$. Concretely, consider a function in Euclidean spaces $f : \mathbb{R}^n \to \mathbb{R}^m$. At any point $\mathbf{a} \in \mathbb{R}^n$, the Fréchet derivative $\partial f$ can be seen as the *directional derivative* of $f$:

$$\partial f : \mathbb{R}^n \to \mathrm{L}(\mathbb{R}^n, \mathbb{R}^m), \quad \partial f(\mathbf{a})(\mathbf{v}) = J_f(\mathbf{a})\mathbf{v} \tag{29}$$

where $J_f(\mathbf{a}) \in \mathbb{R}^{m \times n}$ denote the Jacobian of $f$ at point $\mathbf{a}$ called the *primal*, and $\mathbf{v} \in \mathbb{R}^n$, also called the *tangent* is a vector representing the direction. Therefore the Fréchet derivative is also called *Jacobian-vector-product (JVP)*. And we can write the truncated Taylor expansion as

$$f(\mathbf{a} + \Delta \mathbf{x}) = f(\mathbf{a}) + \partial f(\mathbf{a}, \Delta \mathbf{x}) + \mathcal{O}(\Delta \mathbf{x}^2). \tag{30}$$

Many operators have efficient JVP implementation due to sparsity. For example, element-wise application of scalar function (e.g. activation in neural networks) has diagonal Jacobian, and its JVP can be efficiently implemented as a Hadamard product. Another prominent example is discrete convolution, whose JVP has efficient implementation via FFT.

## B.4 Adjoint of the Fréchet derivative

Given two topological vector spaces $X, Y$, the linear map $u : X \to Y$ has an adjoint ${}^t u : Y' \to X'$ where $X', Y'$ are the dual spaces. The adjoint satisfies the following

$$\forall y \in Y', x \in X, \quad \langle {}^t u(y), x \rangle = \langle y, u(x) \rangle \tag{31}$$

In the finite-dimensional case, the dual space is the space of row vectors, and any linear map can be written as $u(\mathbf{x}) = A\mathbf{x}$. One can easily verify that the adjoint is the transpose: ${}^t u(\mathbf{y}^\top) = \mathbf{y}^\top A$. The *adjoint (transpose)* of the Fréchet derivative of $f : \mathbb{R}^n \to \mathbb{R}^m$, denoted as $\partial^\top f$, is thus defined as

$$\partial^\top f : \mathbb{R}^n \to \mathrm{L}(\mathbb{R}^m, \mathbb{R}^n), \quad \partial^\top f(\mathbf{a})(\mathbf{v}) = \mathbf{v}^\top J_f(\mathbf{a}), \quad \mathbf{v} \in \mathbb{R}^m \tag{32}$$

where $\mathbf{v}^\top$ is the *cotangent* which lives in the dual space of the codomain. Note that the adjoint is taken to $\mathbf{v}$ only where $\mathbf{a}$ is kept fixed. $\partial^\top f$ is also called *vector-Jacobian-product (VJP)*.

## C  Why mixed mode AD schemes like the forward-over-backward might not be better than stacked backward mode AD in the case of PINN

In AD literature [9], the second order derivative is recommended to be computed via forward-over-backward AD, i.e., first do a backward mode AD to get the first order derivative, then apply forward mode AD to the first order derivative to obtain the second order derivative. Usually, we will expect that forward-over-backward AD gives better performance in memory usage over stacked backward AD since the outer differential operator has to differentiate a larger computation graph than the inner one, and forward AD has less overhead as explained in section B.2. Essentially, forward-over-backward reverses the arrows in the third row in Fig. 1, therefore reducing the number of sequential computations and also the size of the evaluation trace. However, in the case of PINN, yet another differentiation to the network parameters $\theta$ needs to be taken. So, computing the second-order differential operator here with forward-over-backward AD might not yield any advantage.

## D  Taylor mode AD

### D.1  High-order Fréchet Derivatives

The $k$th order Fréchet derivative of a function $f : \mathbb{R}^n \to \mathbb{R}^m$ at a point $\mathbf{a}$ is the *multi-linear map with $k$ arguments around point $\mathbf{a}$ that best approximates $f$*. For example, when $k = 2$, we have

$$\partial^2 f : \mathbb{R}^n \to \mathrm{L}(\mathbb{R}^n \times \mathbb{R}^n, \mathbb{R}^m), \quad \partial^2 f(\mathbf{a})(\mathbf{v}, \mathbf{v}') = \mathbf{v}^\top H_f(\mathbf{a})\mathbf{v}' = \sum_{j,k} H_f(\mathbf{a})_{i,j,k} v_j v_k' \tag{33}$$

where $H_f(\mathbf{a}) \in \mathbb{R}^{m \times n \times n}$ denote the Hessian of $f$ at point $\mathbf{a}$, and $\mathbf{v}, \mathbf{v}' \in \mathbb{R}^n$. We can now write the second-order truncated Taylor series with it

$$f(\mathbf{a} + \Delta\mathbf{x}) = f(\mathbf{a}) + \partial f(\mathbf{a})(\Delta\mathbf{x}) + \partial^2 f(\mathbf{a})(\Delta\mathbf{x}, \Delta\mathbf{x}) + \mathcal{O}(\Delta\mathbf{x}^3). \tag{34}$$

For the more general case, we have

$$\partial^k f : \mathbb{R}^n \to \mathrm{L}\left(\bigotimes^k \mathbb{R}^n, \mathbb{R}^m\right), \quad \partial^k f(\mathbf{a})(\mathbf{v}^{(1)}, \dots, \mathbf{v}^{(k)}) = \sum_{i_1, \dots, i_k} D_f^k(\mathbf{a})_{i_0, i_1, \dots, i_k} v_{i_1}^{(1)} \cdots v_{i_k}^{(k)} \tag{35}$$

High-order Fréchet derivative can be seen as the best $k$th order polynomial approximation of $f$ by taking all input tangents to be the same $\mathbf{v} \in \mathbb{R}^n$:

$$f(\mathbf{a} + \Delta\mathbf{x}) = f(\mathbf{a}) + \partial f(\mathbf{a})(\mathbf{v}) + \frac{1}{2}\partial^2 f(\mathbf{a})(\mathbf{v}, \mathbf{v}) + \cdots + \frac{1}{k!}\partial^k f(\mathbf{a})(\mathbf{v}^{\otimes k}) + \mathcal{O}(\Delta\mathbf{x}^{k+1}). \tag{36}$$

### D.2  Composition rule for high-order Fréchet derivatives

Next, we derive the higher-order composition rule by repeatedly applying the usual chain rule.

For composition $f(g(x))$ of scalar functions, we can generalize the chain rule for high-order derivatives by iteratively applying the chain rule to lower-order chain rules:

$$\begin{aligned}
\frac{\partial}{\partial x} f(g(x)) &= f^{(1)}(g(x)) \cdot g^{(1)}(x) \\
\frac{\partial^2}{\partial x^2} f(g(x)) &= f^{(1)}(g(x)) \cdot g^{(2)}(x) + f^{(2)}(g(x)) \cdot [g^{(1)}(x)]^2 \\
\frac{\partial^3}{\partial x^3} f(g(x)) &= f^{(1)}(g(x)) \cdot g^{(3)}(x) + 3f^{(2)}(g(x)) \cdot g^{(1)}(x) \cdot g^{(2)}(x) + f^{(3)}(g(x)) \cdot [g^{(1)}(x)]^3
\end{aligned} \tag{37}$$

where we give the example of up to the third order. For arbitrary $k$, the $k$th order derivative of the composition is given by the Faa di Bruno's formula (scalar version)

$$\frac{\partial^k}{\partial x^k} f(g(x)) = \sum_{\substack{(p_1,\ldots,p_k)\in\mathbb{N}^k, \\ \sum_{i=1}^k i\cdot p_i=k}} \frac{k!}{\prod_i^k p_i!(i!)^{p_i}} \cdot (f^{(\sum_{i=1}^n p_i)}\circ g)(x) \cdot \prod_{j=1}^k \left(\frac{1}{j!}g^{(j)}(x)\right)^{p_j}. \quad (38)$$

where the outermost summation is taken over all partitions of the derivative order $k$. Here a *partition* of $k$ is defined as a tuple $(p_1,\ldots,p_k)\in\mathbb{N}^k$ that satisfies

$$\sum_{i=1}^k i\cdot p_i = k. \quad (39)$$

For vector-valued functions $g:\mathbb{R}^n\to\mathbb{R}^m, f:\mathbb{R}^m\to\mathbb{R}^l$, let

$$\mathbf{a}=g(\mathbf{x})\in\mathbb{R}^m, \quad \mathbf{v}^{(1)}=\frac{\partial g(\mathbf{x})}{\partial\mathbf{x}}\in\mathbb{R}^{m\times n},$$
$$\mathbf{v}^{(2)}=\frac{\partial^2 g(\mathbf{x})}{\partial\mathbf{x}^2}\in\mathbb{R}^{m\times n\times n}, \quad \mathbf{v}^{(3)}=\frac{\partial^3 g(\mathbf{x})}{\partial\mathbf{x}^3}\in\mathbb{R}^{m\times n\times n\times n} \quad (40)$$

we can derive the following composition rule similarly

$$\frac{\partial}{\partial\mathbf{x}} f(g(\mathbf{x})) = D_f(\mathbf{a})_{l,m} v^{(1)}_{m,n}\in\mathbb{R}^{l\times n}$$
$$\frac{\partial^2}{\partial\mathbf{x}^2} f(g(\mathbf{x})) = D_f(\mathbf{a})_{l,m} v^{(2)}_{m,n,n'} + D_f^2(\mathbf{a})_{l,m,m'} v^{(1)}_{m,n} v^{(1)}_{m',n'}\in\mathbb{R}^{l\times n\times n}$$
$$\frac{\partial^3}{\partial\mathbf{x}^3} f(g(\mathbf{x})) = D_f(\mathbf{a})_{l,m} v^{(3)}_{m,n,n',n''} \quad (41)$$
$$+3\cdot D_f^2(\mathbf{a})_{l,m,m'} v^{(1)}_{m,n} v^{(2)}_{m',n',n''}$$
$$+D_f^3(\mathbf{a})_{l,m,m',m''} v^{(1)}_{m,n} v^{(1)}_{m',n'} v^{(1)}_{m'',n''}\in\mathbb{R}^{l\times n\times n\times n}$$

where again we give the example of up to the third order, and repeated indexes are summed as in Einstein notation. The general formula is again given by the multivariate version of the Faa di Bruno's formula. Note that in the multivariate version of the Faa di Bruno's formula, it is possible to take a derivative to distinguishable variables, but here we just present the version with indistinguishable input variables. This gives the composition rule for $k$th order total derivative.

The composition of the high-order Fréchet derivative $\partial^k$ is the case of $n=1$, as the contraction with the input tangents $\mathbf{v}^{(i)}\in\mathbb{R}^d$ is the same as composing with a scalar input function $g:\mathbb{R}\to\mathbb{R}^d$ with $\mathbf{v}^{(i)}=D_g^i$. All derivative tensors of $f(g(x))$ can be represented using a $\mathbb{R}^l$ vector, and similarly all derivative tensor $\mathbf{v}^{(i)}$ of $g$ can be represented using a $\mathbb{R}^m$ vector. Then, the above chain rule can be simplified to

$$\frac{\partial}{\partial t} f(g(t)) = D_f(\mathbf{a})_{l,m} v^{(1)}_m\in\mathbb{R}^l$$
$$\frac{\partial^2}{\partial t^2} f(g(t)) = D_f(\mathbf{a})_{l,m} v^{(2)}_m + D_f^2(\mathbf{a})_{l,m,m'} v^{(1)}_m v^{(1)}_{m'}\in\mathbb{R}^l$$
$$\frac{\partial^3}{\partial t^3} f(g(t)) = D_f(\mathbf{a})_{l,m} v^{(3)}_m + 3\cdot D_f^2(\mathbf{a})_{l,m,m'} v^{(1)}_m v^{(2)}_{m'} + D_f^3(\mathbf{a})_{l,m,m',m''} v^{(1)}_m v^{(1)}_{m'} v^{(1)}_{m''}\in\mathbb{R}^l. \quad (42)$$

The Faa di Bruno's formula again gives the general formula for arbitrary derivative order

$$\frac{\partial^k}{\partial t^k} f(g(t)) = \sum_{\substack{(p_1,\ldots,p_k)\in\mathbb{N}^k, \\ \sum_{i=1}^k i\cdot p_i=k}} \frac{k!}{\prod_i^k p_i!(i!)^{p_i}} \cdot D_f^{\sum_{i=1}^k p_i}(\mathbf{a})_{l,m_1,\ldots,m_{\sum_{i=1}^k p_i}} \cdot \prod_{j=1}^k \left(\frac{1}{j!}v^{(j)}_{m_j}\right)^{p_j}\in\mathbb{R}^l. \quad (43)$$

which is written in the perspective of input primal $\mathbf{a}$ and tangents $\mathbf{v}^{(i)}$.

## E  Removing the mixed partial derivatives term from second order semilinear parabolic PDE

$$\frac{1}{2}\operatorname{tr}\left(\sigma(\mathbf{x},t)\sigma(\mathbf{x},t)^{\top}(\mathrm{Hess}_{\mathbf{x}}u)(\mathbf{x},t)\right) = \frac{1}{2}\operatorname{tr}\left(\sigma(\mathbf{x},t)^{\top}(\mathrm{Hess}_{\mathbf{x}}u)(\mathbf{x},t)\sigma(\mathbf{x},t)\right)$$

$$= \frac{1}{2}\sum_{i=0}^{d}\left[\sigma(\mathbf{x},t)^{\top}(\mathrm{Hess}_{\mathbf{x}}u)(\mathbf{x},t)\sigma(\mathbf{x},t)\right]_{i,i}$$

$$= \frac{1}{2}\sum_{i=0}^{d}\mathbf{e}_{i}^{\top}\sigma(\mathbf{x},t)^{\top}(\mathrm{Hess}_{\mathbf{x}}u)(\mathbf{x},t)\sigma(\mathbf{x},t)\mathbf{e}_{i} \tag{44}$$

$$= \frac{1}{2}\sum_{i=0}^{d}\partial^{2}u((\mathbf{x},t),\sigma(\mathbf{x},t)\mathbf{e}_{i},\mathbf{0}^{\top})_{[3]}.$$

## F  Evaluating arbitrary mixed partial derivatives

### F.1  A concrete example

Let's first consider a concrete case. Suppose the domain is $D$-dimensional we want to compute the mixed derivative $\frac{\partial}{\partial x_i^2 \partial x_j}$. The naive approach would be to compute the entire third order derivative tensor $D_f^3$, which is a tensor of shape $D \times D \times D$, then extract the element at index $(j,i,i)$. However note that from Eq. 43, for any $k > 3$, the pushforward of $k$-jet under $\mathrm{d}^k f$ contains contractions of $D_f^3$. Although in the case of $k = 3$, the only contraction of $D_f^3$ is in the $\partial^3 f$:

$$D_f^3(\mathbf{a})_{l,m,m',m''}v_m^{(1)}v_{m'}^{(1)}v_{m''}^{(1)} \tag{45}$$

which can only be used to compute the diagonal or the block diagonal elements, when $k > 3$, we will have a contraction that computes off-diagonal terms, i.e. the mixed partial derivatives. For example, in $\mathrm{d}^4 f$, if all input tangents are set to zero except for $\mathbf{v}^{(1)}$ and $\mathbf{v}^{(2)}$, $\partial^4 f$ becomes:

$$3 \cdot D_f^2(\mathbf{a})_{l,m_1,m_2}v_{m_1}^{(2)}v_{m_2}^{(2)} + 6 \cdot D_f^3(\mathbf{a})_{l,m_1,m_2,m_3}v_{m_1}^{(2)}v_{m_2}^{(1)}v_{m_3}^{(1)} + D_f^4(\mathbf{a})_{l,m_1,m_2,m_3,m_4}v_{m_1}^{(1)}v_{m_2}^{(1)}v_{m_3}^{(1)}v_{m_4}^{(1)}. \tag{46}$$

which contains the contraction of $D_f^3$ that we want:

$$D_f^3(\mathbf{a})_{l,m_1,m_2,m_3}v_{m_1}^{(2)}v_{m_2}^{(1)}v_{m_3}^{(1)}. \tag{47}$$

However, there are extra terms. We can remove them by doing two extract pushforwards. We can compute the desired mixed partial derivative with the following pushforward of standard basis:

$$\frac{\partial}{\partial x_i^2 \partial x_j}u_\theta(\mathbf{x}) = [\partial^4 u_\theta(\mathbf{x})(\mathbf{e}_i,\mathbf{e}_j,\mathbf{0},\mathbf{0}) - \partial^4 u_\theta(\mathbf{x})(\mathbf{e}_i,\mathbf{0},\mathbf{0},\mathbf{0}) - 3\partial^2 u_\theta(\mathbf{x})(\mathbf{e}_j,\mathbf{0})]/6. \tag{48}$$

If we go to higher-order jets, we can use more flexible contractions, and we can compute the mixed derivative with fewer terms to correct, hence less pushforwards. For example, the pushforward of the fifth-order tangent is

$$10 \cdot D_f^3(\mathbf{a})_{l,m_1,m_2,m_3}v_{m_1}^{(3)}v_{m_2}^{(1)}v_{m_3}^{(1)} + D_f^5(\mathbf{a})_{l,m_1,m_2,m_3,m_4,m_5}v_{m_1}^{(1)}v_{m_2}^{(1)}v_{m_3}^{(1)}v_{m_4}^{(1)}v_{m_5}^{(1)}, \tag{49}$$

if all input tangents are set to zero except for $\mathbf{v}^{(1)}$ and $\mathbf{v}^{(3)}$. With this we only need to remove one term:

$$\frac{\partial}{\partial x_i^2 \partial x_j}u_\theta(\mathbf{x}) = [\partial^5 u_\theta(\mathbf{x})(\mathbf{e}_i,\mathbf{0},\mathbf{e}_j,\mathbf{0},\mathbf{0}) - \partial^5 u_\theta(\mathbf{x})(\mathbf{e}_i,\mathbf{0},\mathbf{0},\mathbf{0},\mathbf{0})]/10. \tag{50}$$

Similarly, by going to the seventh-order tangent, we can compute this mixed derivative with only one pushforward. $\mathrm{d}^7 f$ contains $\partial^7$, and when all input tangents are set to zero except for $\mathbf{v}^{(2)}$ and $\mathbf{v}^{(3)}$, $\partial^7$ equals

$$105 \cdot D_f^3(\mathbf{a})_{l,m_1,m_2,m_3}v_{m_1}^{(3)}v_{m_2}^{(2)}v_{m_3}^{(2)} \tag{51}$$

which is the exact contraction we want. With this we have

$$\frac{\partial}{\partial x_i^2 \partial x_j}u_\theta(\mathbf{x}) = \partial^7 u_\theta(\mathbf{x})(\mathbf{0},\mathbf{e}_i,\mathbf{e}_j,\mathbf{0},\mathbf{0},\mathbf{0},\mathbf{0})/105. \tag{52}$$

### F.2 Procedure for finding the right pushforwards for arbitrary mixed partial derivatives

More generally, consider the case where we need to compute arbitrary mixed partial derivative

$$\frac{\partial^{\sum_j^T q_{i_j}}}{\partial x_{i_1}^{q_{i_1}} \dots \partial x_{i_T}^{q_{i_T}}}, \tag{53}$$

where $T$ is the number of different input dimensions in the mixed partial derivative, and $q_{i_t}$ is the order. To compute it with $k$-jet pushforward, one needs to find:

1. a derivative order $k \in \mathbb{N}$,

2. a sparsity pattern for the tangents $\mathbf{v}^{(i)}$ of the input jet, which is defined as the tuple of $T$ integers $J = (j_1, \dots, j_T)$ where $\mathbf{v}^{(j)} = \mathbf{0}$ when $j \notin J$ and $j_t < k$ for all $t \in [1, T]$,

such that when setting

$$p_j = \begin{cases} 0, & j \notin J \\ q_{i_t}, & j = j_t \end{cases}, \tag{54}$$

$(p_1, p_2, \dots, p_k) \in \mathbb{N}^k$ is a partition of $k$ as defined in Eq. 39.

Let's use the concrete example $\frac{\partial}{\partial x_i^2 \partial x_j}$ again. In this case $T = 2$, $q_{i_1} = 2$ and $q_{i_2} = 1$. We demonstrated that this can be computed with one 7-jet pushforward, which is equivalent to setting $J = (2, 3)$, $k = 2j_1 + j_2 = 7$, and the partition $(0, 2, 1, 0, 0, 0, 0)$. The Faa di Bruno's formula (Eq. 43) ensures that the pushforward of the $k$th order tangent contains a contraction that can be used to compute the desired mixed partial derivative.

Furthermore, if there are no other partitions with a sparsity pattern that is the subset of the sparsity pattern of the partition in consideration, there are no extra terms to remove. Intuitively, if a partition has a sparsity pattern that is not a subset, it will vanish when we set the input tangents to zero according to the sparsity pattern of the partition in consideration. To understand this point better, let's look at the concrete example with the 5-jet pushforward demonstrated above. $(2, 0, 1, 0, 0)$ and $(5, 0, 0, 0, 0)$ are both valid partition of $k = 5$, and the sparsity pattern of $(5, 0, 0, 0, 0)$ is the subset of that of $(2, 0, 1, 0, 0)$: $p_1$ are non-zero in both partition. Therefore the pushforward contains extra terms that can be removed with another pushforward. In the example with 7-jet pushforward, no other partition has the sparsity pattern that is the subset of that of the partition $(0, 2, 1, 0, 0, 0, 0)$. This is equivalent to say, $2 + 2 + 3$ is the only way to sum up to 7 when you can only use 2 and 3, which can be verified easily.

With this setup, it is clear why the diagonal terms can always be computed with pushforward of the lowest possible order: $(k, 0, \dots, 0) \in \mathbb{N}^k$ is always a valid partition $k$, and no other partition has sparsity pattern that is a subset of it.

For mixed partial derivatives, the difficulty scales the total order of the operator $\sum_{t=1}^T q_{i_t}$, and $T$ which can be interpreted as the degree of the "off-diagonalness" of the operator. For example, consider the case where $T = 3$ and $q_{i_1} = 3, q_{i_1} = 2, q_{i_1} = 1$. This corresponds to the operator $\frac{\partial}{\partial x_i^3 \partial x_j^2 \partial x_k}$. To avoid overlapping with the diagonal sparsity pattern $(k, 0, \dots, 0)$ and to keep the order of derivative low, one might try $k = 16$ and the partition $(0, 3, 2, 1, 0, \dots) \in \mathbb{N}^{16}$. However, with higher $k$, there is more chance that other partitions will have a subset sparsity pattern. In this case $(0, 8, 0, 0, 0, \dots) \in \mathbb{N}^{16}$ is one such example. One will need to either find all the partitions with subset sparsity pattern and remove them with multiple pushforward, or further increase the derivative order to find a pattern with no extra term.

## G  Further memory reduction via weight sharing in the first layer

When dealing with high-dimensional data, the parameters of the model's first layer in a conventional fully connected network would grow proportionally with the input dimension, resulting in a significant increase in memory requirements and forming a memory bottleneck due to massive model parameters. To address this issue, convolutional networks are often employed in deep learning for images to reduce the number of model parameters. Here, we adopt a similar approach to mitigate the memory cost of model parameters in high-dimensional PDEs, called weight sharing in the first layer.

Denote the input dimension as $d$, which is potentially excessively high, and the hidden dimension of the MLP as $h$, and assume that $d \gg h$. The first layer weight is an $d \times h$ dimensional matrix, whereas all subsequent layers have a weight matrix with a size of only $h \times h$.

By introducing a weight-sharing scheme, one can reduce the redundancy in the parameters in the first layer. Specifically, we perform an additional 1D convolution to the input vectors $\mathbf{x}_i$ before passing the input into the MLP PINN, as in Fig. 4. The 1D convolution has filter size $B$ that divides $D$ and stride size $B$, so the convolution output is non-overlapping, and the number of channels is set to 1.

This weight-sharing scheme reduces the parameters by approximately $\frac{1}{B}$. The number of parameters in the filters is $B \times 1$, and the subsequent fully connected layer will have a weight matrix of size $\frac{d}{B} \times H$. Therefore, the total number of the first layer is reduced from $d \times h$ to only $\frac{d}{B} \times h + B$, and we can see that with a larger block size $B$, we will have fewer parameters, and the reduction factor is approximately $\frac{1}{B}$. More concretely, suppose $d = 10^6, h = 100$ where one million ($10^6$) dimensional problems are also tested experimentally, so the number of parameters in the first layer is $d \times h = 100 \times 10^6$. If we use a block size of $B = 100$, we will reduce the number of parameters to $\frac{d}{B} \times h + B = 10^6 + 100$. If the block size is $B = 10$, the number of parameters will be $\frac{d}{B} \times h + B = 10 \times 10^6 + 10$. In other words, with a larger block size of $B$, we significantly reduce the number of model parameters.

We will demonstrate the memory efficiency and acceleration thanks to weight-sharing in the experimental section.

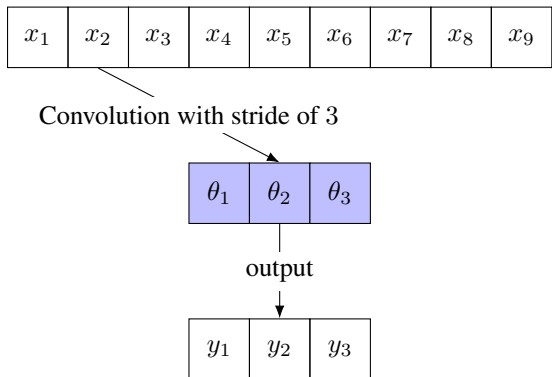

Figure 4: Convolutional weight sharing in the first layer, with input dimension 9 and filter size 3.

## H  Experiment setup

Each experiment is run with five different random seeds, and the average and the standard deviations of these runs are reported.

To get an accurate reading of memory usage, we use a separate run where GPU memory pre-allocation for JAX is disabled through setting the environment variable `XLA_PYTHON_CLIENT_ALLOCATOR=platform`, and the test data set is stored on the CPU memory. The GPU memory usage was obtained via `NVIDIA-smi` and peak memory was reported.

All the experiments were done on a single NVIDIA A100 GPU with 40GB memory and CUDA 12.2. with driver 535.129.03 and JAX version 0.4.23.

**Network architecture and training hyperparameters**    For the semilinear parabolic PDEs tested in Appendix I.2 we follow the network architecture of the original SDGD [13]:

- The network is a 4-layer multi-layer perceptron (MLP) with 128 hidden units activated by Tanh.

- The network is trained with Adam [20] for 10K steps, with an initial learning rate of 1e-3 that linearly decays to 0 in 10K steps, where at each step we calculate the model parameters gradient with 100 uniformly sampled random residual points.

- The model is evaluated using 20K uniformly sampled random points fixed throughout the training.

- The zero boundary condition is satisfied via the following parameterization

$$u_\theta(\mathbf{x}) = (1 - \|\mathbf{x}\|_2^2) u_\theta^{\text{MLP}}(\mathbf{x}) \tag{55}$$

where $u_\theta^{\text{MLP}}$ is the MLP network, and $u_\theta$ is the PDE ansatz, as described in [26].

For the semilinear parabolic PDEs tested in Appendix I.2, we made the following modifications:

- Instead of using re-parameterization, the boundary/initial condition is satisfied by adding a regularization loss to the residual loss:

$$\ell_{\text{boundary}}(\theta; \{\mathbf{x}_{b,i}\}_{i=1}^{N_b}) = \frac{1}{N_b} \sum_{i=1}^{N_b} |u_\theta(\mathbf{x}_{b,i}, 0) - g(\mathbf{x}_{b,i})|^2 + C_g \cdot \frac{1}{N_b} \sum_{i=1}^{N_b} |\nabla u_\theta(\mathbf{x}_{b,i}, 0) - \nabla g(\mathbf{x}_{b,i})|^2 \tag{56}$$

where $g(\cdot)$ is the initial data, $N_b$ is the batch size for boundary points, $u_\theta$ is the PDE ansatz, $C_g$ is the coefficient for the first-order derivative boundary loss term, which we set to 0.05. The total loss is

$$\ell_{\text{residual}}(\theta; \{\mathbf{x}_{r,i}\}_{i=1}^{N_r}) + 20 \ell_{\text{boundary}}(\theta; \{\mathbf{x}_{b,i}\}_{i=1}^{N_b}). \tag{57}$$

- Instead of discretizing the time and sample residual points using the underlying stochastic process, we uniformly sample the time steps between the initial and the terminal time, i.e. $t \sim \text{uniform}[0, T]$, and then sample $\mathbf{x}$ directly from the distribution of $\mathbf{X}_t$, i.e. $\mathbf{x} \sim \mathcal{N}(0, (T-t) \cdot \mathbf{I}_{d \times d})$. To match the original training setting of 100 SDE trajectories with 0.015 step size for time discretization, we use a batch size of 2000 for residual points and 100 for boundary/initial points.

- We use a 4-layer multi-layer perceptron (MLP) with 1024 hidden units activated by Tanh. The network is trained with Adam [20] for 10K steps, with an initial learning rate of 1e-3 that exponentially decays with exponent 0.9995.

- To test the quality of the PINN solution, we measure the relative L1 error at the point $(\mathbf{x}_{\text{test}}, T)$ against the reference value computed via multilevel Picard's method [3, 4, 17].

In all experiments, we use the biased version of Eq. 25:

$$\tilde{\ell}_{\text{residual}}(\theta; \{\mathbf{x}^{(i)}\}_{i=1}^{N_r}, J) = \frac{1}{N_r} \sum_{i=1}^{N_r} \left[ \tilde{\mathcal{L}}_J u_\theta(\mathbf{x}^{(i)}) - f(\mathbf{x}^{(i)}) \right] \tag{58}$$

as the bias in practice is very small and does not affect convergence.

# I  Experiments Results

## I.1  Inseparable and effectively high-dimensional PDEs

The first class of PDEs is defined via a nonlinear, inseparable, and effectively high-dimensional exact solution $u_{\text{exact}}(\mathbf{x})$ defined within the $d$-dimensional unit ball $\mathbb{B}^d$:

$$\begin{aligned} \mathcal{L}u(\mathbf{x}) &= f(\mathbf{x}), \quad \mathbf{x} \in \mathbb{B}^d \\ u(\mathbf{x}) &= 0, \qquad \mathbf{x} \in \partial\mathbb{B}^d \end{aligned} \tag{59}$$

where $\mathcal{L}$ is a linear/nonlinear operator and $g(\mathbf{x}) = \mathcal{L}u_{\text{exact}}(\mathbf{x})$. The zero boundary condition ensures that no information about the exact solution is leaked through the boundary condition. We will consider the following operators:

- Poisson equation: $\mathcal{L}u(\mathbf{x}) = \nabla^2 u(\mathbf{x})$.
- Allen-Cahn equation: $\mathcal{L}u(\mathbf{x}) = \nabla^2 u(\mathbf{x}) + u(\mathbf{x}) - u(\mathbf{x})^3$.
- Sine-Gordon equation: $\mathcal{L}u(\mathbf{x}) = \nabla^2 u(\mathbf{x}) + \sin(u(\mathbf{x}))$.

For the exact solution, we consider the following with all $c_i \sim \mathcal{N}(0, 1)$:

- two-body interaction: $u_{\text{exact}}(\mathbf{x}) = (1 - \|\mathbf{x}\|_2^2) \left( \sum_{i=1}^{d-1} c_i \sin(x_i + \cos(x_{i+1}) + x_{i+1} \cos(x_i)) \right)$.

- three-body interaction: $u_{\text{exact}}(\mathbf{x}) = (1 - \|\mathbf{x}\|_2^2) \left( \sum_{i=1}^{d-2} c_i \exp(x_i x_{i+1} x_{i+2}) \right)$.

We tested the performance of STDE on these equations, and the results are presented in Table 3, 4, 5, 6. For the Allen-Cahn equation, we performed a detailed ablation study (Table 3), and we expect these results to generalize over these second-order PDEs.

### I.1.1 Further details on ablation study

**The gain by using JAX instead of PyTorch** Since the original SDGD was implemented in PyTorch, we implemented the stacked backward mode without parallelization in SDGD dimensions in JAX for fair comparison (dubbed as "Stacked Backward mode SDGD in JAX" in Table 3). The for-loop over SDGD dimension is implemented using `jax.lax.scan`. Table 3 shows that, even with the original stacked backward mode AD, the speed of JAX implementation can be more than $10\times$ faster when the dimension is high. The memory profile is similar. The difference could come from the fact that JAX uses XLA to perform Just-in-time (JIT) compilation of the Python code into optimized kernels. However, note that for the case of 100,000 dimensions, the JAX implementation of the stacked backward mode AD encountered an out-of-memory (OOM) error. This is because performing JIT compilation requires extra memory, and the peak memory requirement during JIT compilation is higher than that during training.

**Randomization batch size** We also tested the case where the STDE randomization batch size is reduced to 16. As seen in Table 3, in the case of Allen-Cahn provides $\sim 2\times$ speed up, without hurting performance. However, theoretically lowering the randomization batch size leads to higher variance. The trade-off between computational efficiency and stability in convergence warrants further studies.

### I.2 Semilinear Parabolic PDEs

The second class of PDEs is the semilinear parabolic PDEs, where the initial condition is specified:

$$\begin{aligned}
\frac{\partial}{\partial t}u(\mathbf{x},t) &= \mathcal{L}u(\mathbf{x},t) \quad (\mathbf{x},t) \in \mathbb{R}^d \times [0,T] \\
u(\mathbf{x},t) &= g(\mathbf{x}), \quad (\mathbf{x},t) \in \mathbb{R}^d \times \{0\}
\end{aligned} \tag{60}$$

where $g(\mathbf{x})$ is a known, analytical, and time-independent function that specifies the initial condition, and $T$ is the terminal time. We aim to approximate the solution's true value at one test point $\mathbf{x}_{\text{test}} \in \mathbb{R}^d$, at the terminal time $t = T$, i.e. at $(\mathbf{x}_{\text{test}}, T)$.

We will consider the following operators

- Semilinear Heat Eq.

$$\mathcal{L}u(\mathbf{x},t) = \nabla^2 u(\mathbf{x},t) + \frac{1 - u(\mathbf{x},t)^2}{1 + u(\mathbf{x},t)^2}. \tag{61}$$

  with initial condition $g(\mathbf{x}) = 5/(10 + 2\|\mathbf{x}\|^2)$,

- Allen-Cahn equation

$$\mathcal{L}u(\mathbf{x},t) = \nabla^2 u(\mathbf{x},t) + u(\mathbf{x},t) - u(\mathbf{x},t)^3. \tag{62}$$

  with initial condition $g(\mathbf{x}) = \arctan(\max_i x_i)$,

- Sine-Gordon equation

$$\mathcal{L}u(\mathbf{x},t) = \nabla^2 u(\mathbf{x},t) + \sin(u(\mathbf{x},t)). \tag{63}$$

  with initial condition $g(\mathbf{x}) = 5/(10 + 2\|\mathbf{x}\|^2)$,

All three equation uses the test point $\mathbf{x}_{\text{test}} = \mathbf{0}$ and terminal time $T = 0.3$.

### I.3 Weight sharing

We tested the weight-sharing technique mentioned in Section G.

In this section, we evaluate the performance of the weight-sharing scheme described in Appendix G. We tested the best-performing method from Table 3 (STDE with small randomization batch size of 16) with different weight-sharing block sizes, on the inseparable Allen-Cahn equation with the two-body exact solution.

Table 3: Computational results for the Inseparable Allen-Cahn equation with the two-body exact solution, where the randomization batch size is set to 100 unless stated otherwise.

| Method | Metric | 100 D | 1K D | 10K D | 100K D | 1M D |
|---|---|---|---|---|---|---|
| Backward mode SDGD (PyTorch) [13] | Speed | 55.56it/s | 3.70it/s | 1.85it/s | 0.23it/s | OOM |
| | Memory | 1328MB | 1788MB | 4527MB | 32777MB | OOM |
| | Error | 7.187E-03 | 5.615E-04 | 1.864E-03 | 2.178E-03 | OOM |
| Backward mode SDGD (JAX) | Speed | 40.63it/s | 37.04it/s | 29.85it/s | OOM | OOM |
| | Memory | 553MB | 565MB | 1217MB | OOM | OOM |
| | Error | 3.51E-03 ±8.47E-05 | 7.29E-04 ±5.45E-06 | 3.46E-03 ±2.01E-04 | OOM | OOM |
| Parallelized backward mode SDGD | Speed | 1376.84it/s | 845.21it/s | 216.83it/s | 29.24it/s | OOM |
| | Memory | 539MB | 579MB | 1177MB | 4931MB | OOM |
| | Error | 6.87E-03 ±6.97E-05 | 3.12E-03 ±7.04E-04 | 2.59E-03 ±2.20E-05 | 1.60E-03 ±1.13E-05 | OOM |
| Forward-over -Backward SDGD | Speed | 778.18it/s | 560.91it/s | 193.91it/s | 27.18it/s | OOM |
| | Memory | 537MB | 579MB | 1519MB | 4929MB | OOM |
| | Error | 4.07E-03 ±7.42E-05 | 2.19E-03 ±2.03E-04 | 5.47E-04 ±7.48E-05 | 4.21E-03 ±2.53E-04 | OOM |
| Forward Laplacian [24] | Speed | **1974.50it/s** | 373.73it/s | 32.15it/s | OOM | OOM |
| | Memory | 507MB | 913MB | 5505MB | OOM | OOM |
| | Error | 4.33E-03 ±4.97E-05 | 5.50E-04 ±4.60E-05 | 5.58E-03 ±2.73E-04 | OOM | OOM |
| STDE | Speed | 1035.09it/s | 1054.39it/s | 454.16it/s | 156.90it/s | 13.61it/s |
| | Memory | 543MB | 537MB | 795MB | 1073MB | **6235MB** |
| | Error | 1.03E-02 ±7.69E-05 | 6.21E-04 ±2.22E-04 | 3.45E-03 ±1.17E-05 | 2.59E-03 ±7.93E-06 | 1.38E-03 ±3.34E-05 |
| STDE (batch size=16) | Speed | 1833.78it/s | **1559.36it/s** | **587.60it/s** | **283.33it/s** | **21.34it/s** |
| | Memory | **457MB** | **481MB** | **741MB** | 1063MB | 6295MB |
| | Error | 1.89E-02 ±2.37E-04 | 7.07E-04 ±1.02E-05 | 8.33E-04 ±2.96E-04 | 1.50E-03 ±1.02E-05 | 3.99E-03 ±3.41E-05 |

Table 4: Computational results for the Inseparable Poisson equation with two-body exact solution.

| Method | Metric | 100D | 1K D | 10K D | 100K D | 1M D |
|---|---|---|---|---|---|---|
| Backward mode SDGD (PyTorch) [13] | Speed | 55.56it/s | 3.70it/s | 1.85it/s | 0.23it/s | OOM |
| | Memory | 1328MB | 1788MB | 4527MB | 32777MB | OOM |
| | Error | 7.189E-03 | 5.611E-04 | 1.850E-03 | 2.175E-03 | OOM |
| STDE (batch size=16) | Speed | 2020.05it/s | 1649.20it/s | 584.98it/s | 281.78it/s | 20.38it/s |
| | Memory | 457MB | 481MB | 741MB | 1063MB | 6295MB |
| | Error | 3.50E-03 ±1.44E-04 | 4.91E-04 ±3.45E-05 | 4.70E-03 ±2.10E-05 | 3.49E-03 ±2.14E-05 | 9.18E-04 ±6.39E-06 |

Table 5: Computational results for the Inseparable Sine-Gordon equation with two-body exact solution.

| Method | Metric | 100 D | 1K D | 10K D | 100K D | 1M D |
|---|---|---|---|---|---|---|
| Backward mode SDGD (PyTorch) [13] | Speed | 55.56it/s | 3.70it/s | 1.85it/s | 0.23it/s | OOM |
| | Memory | 1328MB | 1788MB | 4527MB | 32777MB | OOM |
| | Error | 7.192E-03 | 5.641E-04 | 1.854E-03 | 2.177E-03 | OOM |
| STDE (batch size=16) | Speed | 1926.33it/s | 1467.38it/s | 566.26it/s | 279.24it/s | 19.88it/s |
| | Memory | 457MB | 481MB | 741MB | 1063MB | 6295MB |
| | Error | 3.64E-03 ±1.46E-04 | 5.40E-04 ±7.21E-05 | 5.32E-03 ±5.12E-04 | 9.56E-04 ±8.03E-06 | 9.47E-04 ±8.30E-06 |

Table 6: Computational results for the Inseparable Allen-Cahn, Poisson, and Sine-Gordon equation with the three-body exact solution, computed via STDE with randomization batch size $|J|$ set to 16. *STDE with randomization batch size ($|J|$) of 16 performs poorly on the 1M dimensional Inseparable Poisson equation with three-body exact solution: the L2 relative error is only 9.05E-02± 6.88E-04. To get better convergence, we increase the randomization batch size to 50 for the 1M case. This incurs no extra memory cost and is only slightly slower than the original setting (speed is 46.80it/s when randomization batch size is 16).

| Eq. | Metric | 100 D | 1K D | 10K D | 100K D | 1M D |
|---|---|---|---|---|---|---|
| Allen-Cahn | Speed | 1938.80it/s | 1840.21it/s | 1291.67it/s | 356.76it/s | 46.97it/s |
| | Memory | 461MB | 481MB | 539MB | 1055MB | 6233MB |
| | Error | 9.97E-03 ±3.89E-04 | 1.43E-03 ±1.60E-04 | 6.21E-04 ±8.15E-05 | 1.56E-05 ±3.28E-07 | 2.25E-06 ±1.48E-07 |
| Poisson * | Speed | 1991.28it/s | 1872.31it/s | 1276.21it/s | 364.04it/s | 31.73it/s |
| | Memory | 473MB | 481MB | 539MB | 1055MB | 6233MB |
| | Error | 1.00E-02 ±3.27E-04 | 1.02E-03 ±3.67E-05 | 1.01E-04 ±2.40E-07 | 9.26E-02 ±5.36E-04 | 4.82E-06 ±2.16E-07 |
| Sine-Gordon | Speed | 1938.80it/s | 1840.21it/s | 1291.67it/s | 356.76it/s | 46.88it/s |
| | Memory | 475MB | 479MB | 539MB | 1063MB | 6233MB |
| | Error | 9.97E-03 ±3.89E-04 | 1.43E-03 ±1.60E-04 | 6.21E-04 ±8.15E-05 | 1.56E-05 ±3.28E-07 | 2.31E-05 ±1.48E-06 |

Table 7: Computational results for the Time-dependent Semilinear Heat equation, where the number of SDGD sampled dimensions is set to 10.

| Method | Metric | 10 D | 100 D | 1K D | 10K D |
|---|---|---|---|---|---|
| Backward mode SDGD (PyTorch) [13] | Speed | - | - | - | - |
| | Memory | - | - | - | - |
| | Error | 1.052E-03 | 5.263E-04 | 6.910E-03 | 1.598E-03 |
| BackwardBackward mode SDGD (JAX) | Speed | 211.63it/s | 207.66it/s | 188.31it/s | 93.21it/s |
| | Memory | **619MB** | **621MB** | **655MB** | 1371MB |
| | Error | 8.55E-05 ±6.75E-05 | 4.02E-04 ±2.07E-04 | 3.81E-04 ±4.43E-04 | 2.60E-03 ±1.38E-03 |
| STDE | Speed | **660.82it/s** | **635.16it/s** | **599.15it/s** | **361.11it/s** |
| | Memory | 625MB | 625MB | 657MB | **971MB** |
| | Error | 6.99E-05 ±5.78E-05 | 3.69E-04 ±2.19E-04 | 3.38E-04 ±3.30E-04 | 6.08E-03 ±7.47E-03 |

Table 8: Computational results for the Time-dependent Allen-Cahn equation, where the number of SDGD sampled dimensions is set to 10.

| Method | Metric | 10 D | 100 D | 1K D | 10K D |
|---|---|---|---|---|---|
| BackwardBackward mode SDGD (PyTorch) [13] | Speed | - | - | - | - |
| | Memory | - | - | - | - |
| | Error | 7.815E-04 | 3.142E-04 | 7.042E-04 | 2.477E-04 |
| Backward mode SDGD (JAX) | Speed | 211.38it/s | 206.42it/s | 188.02it/s | 93.20it/s |
| | Memory | 619MB | 621MB | 657MB | 1371MB |
| | Error | 6.31E-02 ±3.79E-02 | 4.38E-03 ±2.48E-03 | 1.35E-03 ±1.23E-03 | 3.97E-04 ±3.03E-04 |
| STDE | Speed | **677.51it/s** | **650.98it/s** | **598.33it/s** | **361.31it/s** |
| | Memory | **533MB** | **535MB** | 657MB | **903MB** |
| | Error | 6.37E-02 ±3.77E-02 | 4.38E-03 ±2.47E-03 | 1.26E-03 ±1.29E-03 | 3.79E-04 ±2.75E-04 |

Table 9: Computational results for the Time-dependent Sine-Gordon equation, where the number of SDGD sampled dimensions is set to 10.

| Method | Metric | 10 D | 100 D | 1K D | 10K D |
|---|---|---|---|---|---|
| BackwardBackward mode SDGD (PyTorch) [13] | Speed | - | - | - | - |
| | Memory | - | - | - | - |
| | Error | 7.815E-04 | 3.142E-04 | 7.042E-04 | 2.477E-04 |
| BackwardBackward mode SDGD (JAX) | Speed | 210.83it/s | 207.44it/s | 187.98it/s | 93.17it/s |
| | Memory | 619MB | 621MB | 655MB | 1371MB |
| | Error | 5.39E-05 ±4.10E-05 | 9.15E-05 ±6.06E-05 | 4.19E-04 ±2.18E-04 | 3.74E-02 ±4.15E-02 |
| STDE | Speed | 629.04it/s | 608.83it/s | 596.12it/s | 365.09it/s |
| | Memory | 525MB | 539MB | 655MB | 971MB |
| | Error | 4.15E-05 ±3.21E-05 | 2.54E-04 ±1.76E-04 | 4.05E-03 ±1.44E-02 | 1.66E-02 ±5.95E-03 |

From Table 10, we can see that weight sharing drastically reduces the number of network parameters and memory usage. With $B = 50$, there is a 2.5x reduction in memory and there is no performance loss in terms of L2 relative error.

However, from the experiments we can see that, in both the 1M and the 5M case, increasing the block size beyond 50 provides diminishing returns. For the 1M case, increasing $B$ to 1000 affects the convergence quality, as the L2 relative error goes up by 100x. For 5M, the maximum block size one can use before degrading performance is 500, which is expected as the dimensionality of the problem is higher.

From Table 10 we can also see that in the 5M-dimensional case, we will have an out-of-memory (OOM) error without weight sharing. With weight sharing enabled, we can effectively solve the 5M-dimensional PDE with good relative L2 error, in around 30 minutes.

## I.4    High-order PDEs

Here we demonstrate how to use STDE to calculate mixed partial derivatives in some actual PDE. We will consider the 2D Korteweg-de Vries (KdV) equation and the 2D Kadomtsev-Petviashvili equation from [32], and the regular 1D KdV equation with gPINN [42].

Table 10: Effects of different weight sharing block sizes $B$ for the Inseparable Allen-Cahn equation with two-body exact solution solved with STDE with randomization batch size of 16. $B = 1$ equals no weight sharing.

| dim | | $B = 1$ | $B = 10$ | $B = 50$ | $B = 100$ | $B = 500$ | $B = 1000$ |
|-----|------|---------|----------|----------|-----------|-----------|------------|
| 1M | Speed | 21.34it/s | 16.67it/s | 23.14it/s | 23.73it/s | 25.47it/s | 26.60it/s |
| | Memory | 6295MB | 4819MB | 2505MB | 2461MB | 2409MB | 2403MB |
| | #Param. | 128,033,281 | 12,833,292 | 2,593,332 | 1,313,382 | 289,782 | 162,282 |
| | Error | 3.99E-03 $\pm$3.41E-05 | 1.86E-02 $\pm$3.13E-04 | 4.76E-03 $\pm$1.27E-04 | 1.22E-03 $\pm$6.05E-05 | 2.57E-03 $\pm$1.15E-04 | 6.06E-01 $\pm$4.17E-04 |
| 5M | Speed | OOM | 3.16it/s | 4.47it/s | 4.74it/s | 4.82it/s | 4.76it/s |
| | Memory | OOM | 25023MB | 10595MB | 10359MB | 10163MB | 10143MB |
| | #Param. | 640,033,281 | 64,033,292 | 12,833,332 | 6,433,382 | 1,313,782 | 674,282 |
| | Error | OOM | 5.11E-01 $\pm$4.01E-04 | 3.13E-03 $\pm$2.34E-04 | 3.94E-03 $\pm$2.22E-04 | 1.98E-03 $\pm$5.20E-05 | 6.27E-01 $\pm$3.03E-04 |

We will demonstrate that STDE increases the speed for computing the mixed partial derivatives, as it avoids computing the entire derivative tensor. Since these equations are low-dimensional we do not need to sample over the space dimension.

In this section, the equations are all time-dependent and the space is 2D, and we will omit the argument to the solution, i.e. we will write $u(\mathbf{x}, t) = u$. To test the speed improvement, we run the STDE implementation against repeated backward mode AD on a Nvidia A100 GPU with 40GB memory. The results are reported in Table 11. From the Table we see that STDE provides around $\sim 2\times$ speed up compared to repeated application of backward mode AD across different network sizes.

### I.4.1 High-order low-dimensional PDEs

**Alternative way to compute the terms in 2D Korteweg-de Vries (KdV) equation** The terms in the 2D KdV equation

$$u_{ty} + u_{xxxy} + 3(u_y u_x)_x - u_{xx} + 2u_{yy} = 0. \tag{64}$$

can alternatively be computed with the pushforward of the following jets

$$\mathfrak{J}^{(1)} = \mathrm{d}^9 u(\mathbf{x}, \mathbf{0}, \mathbf{e}_x, \mathbf{e}_y, \mathbf{0}, \dots), \ \ \mathfrak{J}^{(2)} = \mathrm{d}^3 u(\mathbf{x}, \mathbf{0}, \mathbf{e}_y, \mathbf{e}_t), \ \ \mathfrak{J}^{(3)} = \mathrm{d}^3 u(\mathbf{x}, \mathbf{0}, \mathbf{e}_y, \mathbf{0}). \tag{65}$$

All the derivative terms can be found in these output jets $\{\mathfrak{J}^{(i)}\}$:

$$u_x = \mathfrak{J}^{(1)}_{[2]}, \ u_y = \mathfrak{J}^{(1)}_{[3]}, \ u_{xx} = \mathfrak{J}^{(1)}_{[4]}/3, \ u_{xy} = \mathfrak{J}^{(1)}_{[5]}/10, u_{yy} = \mathfrak{J}^{(3)}_{[2]},$$
$$u_{yyy} = \mathfrak{J}^{(3)}_{[3]}, u_{xxxy} = (\mathfrak{J}^{(1)}_{[9]} - 280 u_{yyy})/840, \ u_{ty} = (\mathfrak{J}^{(2)}_{[3]} - u_{yyy})/3, \tag{66}$$

**2D Kadomtsev-Petviashvili (KP) equation** Consider the following equation

$$(u_t + 6u u_x + u_{xxx})_x + 3\sigma^2 u_{yy} = 0. \tag{67}$$

which can be expanded as

$$u_{tx} + 6u_x u_x + 6u u_{xx} + u_{xxxx} + 3\sigma^2 u_{yy} = 0. \tag{68}$$

All the derivative terms can be computed with a 5-jet, 4-jet, and a 2-jet pushforward. Let

$$\mathfrak{J}^{(1)} := \mathrm{d}^5 u(\mathbf{x}, \mathbf{0}, \mathbf{e}_t, \mathbf{e}_x, \mathbf{0}, \mathbf{0})$$
$$\mathfrak{J}^{(2)} := \mathrm{d}^4 u(\mathbf{x}, \mathbf{e}_x, \mathbf{0}, \mathbf{0}, \mathbf{0}) \tag{69}$$
$$\mathfrak{J}^{(3)} := \mathrm{d}^2 u(\mathbf{x}, \mathbf{e}_y, \mathbf{0}).$$

Then all required derivative terms can be evaluated as follows.

$$u_{tx} = \mathfrak{J}^{(1)}_{[5]}/10,$$
$$u_x = \mathfrak{J}^{(2)}_{[1]}, \ u_{xx} = \mathfrak{J}^{(2)}_{[2]}, \ u_{xxxx} = \mathfrak{J}^{(2)}_{[4]}, \tag{70}$$
$$u_{yy} = \mathfrak{J}^{(3)}_{[2]}.$$

**Gradient-enhanced 1D Korteweg-de Vries (g-KdV) equation** Consider the following equation

$$u_t + uu_x + \alpha u_{xxx} = 0. \tag{71}$$

Gradient-enhanced PINN (gPINN) [42] regularizes the learned PINN such that the gradient of the residual is close to the zero vector. This increases the accuracy of the solution. Specifically, the PINN loss (Eq. 24) is augmented with the term

$$\ell_{\text{gPINN}} \left( \{ \mathbf{x}^{(i)} \}_{i=1}^{N_r} \right) = \frac{1}{N_r} \sum_i \sum_j^d \left| \frac{\partial}{\partial x_j} R(\mathbf{x}^{(i)}) \right|^2. \tag{72}$$

The total loss becomes

$$\ell_{\text{residual}} + c_{\text{gPINN}} \, \ell_{\text{gPINN}} \tag{73}$$

where $c_{\text{gPINN}}$ is the g-PINN penalty weight. To perform gradient-enhancement we need to compute the gradient of the residual:

$$R(x,t) := u_t + uu_x + \alpha u_{xxx},$$
$$\nabla R(x,t) = [u_{tt} + u_t u_x + uu_{tx} + \alpha u_{txxx}, \quad u_{tx} + u_x u_x + uu_{xx} + \alpha u_{xxxx}]. \tag{74}$$

All the derivative terms can be computed with one 2-jet and two 7-jet pushforward. Let

$$\mathfrak{J}^{(1)} := \mathrm{d}^7 u(\mathbf{x}, \mathbf{e}_x, \mathbf{0}, \mathbf{0}, \mathbf{0}, \mathbf{0}, \mathbf{0}, \mathbf{0})$$
$$\mathfrak{J}^{(2)} := \mathrm{d}^7 u(\mathbf{x}, \mathbf{e}_x, \mathbf{0}, \mathbf{0}, \mathbf{e}_t, \mathbf{0}, \mathbf{0}, \mathbf{0}) \tag{75}$$
$$\mathfrak{J}^{(3)} := \mathrm{d}^2 u(\mathbf{x}, \mathbf{e}_t, \mathbf{0}).$$

Then all required derivative terms can be evaluated as follows.

$$u_x = \mathfrak{J}^{(1)}_{[1]}, \ u_{xx} = \mathfrak{J}^{(1)}_{[2]}, \ u_{xxx} = \mathfrak{J}^{(1)}_{[3]}, \ u_{xxxx} = \mathfrak{J}^{(1)}_{[4]}, \ u_{xxxxx} = \mathfrak{J}^{(1)}_{[5]},$$
$$u_{txxx} = (\mathfrak{J}^{(2)}_{[7]} - \mathfrak{J}^{(1)}_{[8]})/35, \ u_{tx} = (\mathfrak{J}^{(2)}_{[5]} - u_{xxxxx})/5, \ u_t = \mathfrak{J}^{(2)}_{[4]} - u_{xxxx}, \tag{76}$$
$$u_{tt} = \mathfrak{J}^{(3)}_{[2]}.$$

Table 11: Speed scaling for training low-dimensional high-order PDEs with different network sizes. The base network has depth $L = 4$ and width $h = 128$. STDE* is the alternative scheme using lower-order pushforwards.

| Speed (it/s) ↑ | network size | Base | $L = 8$ | $L = 16$ | $h = 256$ | $h = 512$ | $h = 1024$ |
|---|---|---|---|---|---|---|---|
| | Backward | 762.86 | 279.19 | 123.20 | 656.01 | 541.10 | 349.23 |
| 2D KdV | STDE | 1372.41 | 642.82 | 303.39 | 1209.30 | 743.75 | 418.13 |
| | STDE* | 1357.64 | 606.43 | 272.01 | 1203.97 | 841.07 | 442.32 |
| 2D KP | Backward | 766.79 | 278.53 | 123.67 | 642.34 | 525.23 | 340.94 |
| | STDE | 1518.82 | 676.16 | 304.95 | 1498.61 | 1052.62 | 642.21 |
| 1D g-KdV | Backward | 621.04 | 232.35 | 102.39 | 559.65 | 482.52 | 293.97 |
| | STDE | 1307.27 | 593.21 | 253.48 | 1187.31 | 776.65 | 441.50 |

### I.4.2 Amortized gradient-enhanced PINN for high-dimensional PDEs

It is expensive to apply gradient enhancement for high-dimensional PDEs. For example, the gradient of the residual for the inseparable Allen-Cahn equation described in I.1 is given by

$$\frac{\partial}{\partial x_j} R(\mathbf{x}) = \frac{\partial}{\partial x_j} \left[ \sum_i \frac{\partial^2}{\partial x_i^2} u(\mathbf{x}) + u(\mathbf{x}) - u^3(\mathbf{x}) - f(\mathbf{x}) \right]$$
$$= \sum_{i=1}^d \frac{\partial^3}{\partial x_j \partial x_i^2} u(\mathbf{x}) + \frac{\partial}{\partial x_j} u(\mathbf{x}) - 3u^2(\mathbf{x}) \frac{\partial}{\partial x_j} u(\mathbf{x}) - \frac{\partial}{\partial x_j} f(\mathbf{x}). \tag{77}$$

With STDE randomization, we randomized the second order term $\frac{\partial^2}{\partial x_i^2}$ with index $i$ sampled from $[1,d]$. We can also sample the gPINN penalty terms. As mentioned in Appendix F.1, we have

$$\mathfrak{J} = \mathrm{d}^7 u(\mathbf{x}, \mathbf{0}, \mathbf{e}_i, \mathbf{e}_j, \mathbf{0}, \mathbf{0}, \mathbf{0}, \mathbf{0}), \quad \frac{\partial}{\partial x_i^2 \partial x_j} u(\mathbf{x}) = \mathfrak{J}_{[7]}/105. \tag{78}$$

We further have

$$\frac{\partial^2}{\partial x_i^2} u(\mathbf{x}) = \mathfrak{J}_{[4]}/3, \tag{79}$$

so the STDE of the Laplacian operator can be computed together with the above pushforward. With this pushforward, we can efficiently amortize the gPINN regularization loss by minimizing the following upperbound on the original gPINN loss with randomized Laplacian

$$\tilde{\ell}_{\text{gPINN}} \left( \{\mathbf{x}^{(i)}\}_{i=1}^{N_r}, I, J \right)$$

$$= \frac{1}{N_r} \sum_{j \in J} \sum_{i \in I} \left| \frac{\partial^3}{\partial x_j \partial x_i^2} u(\mathbf{x}) + \frac{\partial}{\partial x_j} u(\mathbf{x}) - 3u^2(\mathbf{x}) \frac{\partial}{\partial x_j} u(\mathbf{x}) - \frac{\partial}{\partial x_j} f(\mathbf{x}) \right|^2$$

$$\geq \frac{1}{N_r} \sum_{j \in J} \left| \sum_{i \in I} \frac{\partial^3}{\partial x_j \partial x_i^2} u(\mathbf{x}) + \frac{\partial}{\partial x_j} u(\mathbf{x}) - 3u^2(\mathbf{x}) \frac{\partial}{\partial x_j} u(\mathbf{x}) - \frac{\partial}{\partial x_j} f(\mathbf{x}) \right|^2, \tag{80}$$

where $J$ is an independently sampled index set for sampling the gPINN terms. The total loss is

$$\tilde{\ell}_{\text{residual}}(\theta; \{\mathbf{x}^{(i)}\}_{i=1}^{N_r}, I) + \tilde{\ell}_{\text{gPINN}} \left( \{\mathbf{x}^{(i)}\}_{i=1}^{N_r}, I, J \right). \tag{81}$$

We call this technique **amortized gPINN**. The above formula applies to all PDEs where the derivative operator is the Laplacian. For example, for the Sine-Gordon equation, we have

$$\tilde{\ell}_{\text{gPINN}} \left( \{\mathbf{x}^{(i)}\}_{i=1}^{N_r}, I, J \right)$$

$$= \frac{1}{N_r} \sum_{j \in J} \sum_{i \in I} \left| \frac{\partial^3}{\partial x_j \partial x_i^2} u(\mathbf{x}) + \cos u(\mathbf{x}) \frac{\partial}{\partial x_j} u(\mathbf{x}) - \frac{\partial}{\partial x_j} f(\mathbf{x}) \right|^2. \tag{82}$$

We use $c_{\text{gPINN}} = 0.1$, and to get better convergence, we train for $20K$ steps instead of $10K$ steps as in all other experiments in this paper. The results are reported in Table 12. We implement the baseline method based on the best performing first-order AD scheme, the parallelized backward mode SDGD, which we denoted as JVP-HVP in the table. Specifically, to compute the residual gradient we apply one more JVP to the HVP-based implementation of Laplacian (Appendix A.2). From the table, we see that STDE-based amortized gPINN performs better than the JVP-HVP implementation, and both are more efficient than applying backward mode AD in a for-loop. Furthermore, through amortizing we can apply gPINN to high-dimensional PDE which was intractable.

## J   Pushing forward dense random jets

In this section we establish the connection between the classical technique of HTE [16] and STDE by demonstrating that HTE is a pushforward of dense isotropic random 2-jet.

### J.1   Review of HTE

HTE provides a random estimation of the trace of a matrix $A \in \mathbb{R}^{d \times d}$ as follows:

$$\mathrm{tr}(A) = \mathbb{E}_{\mathbf{v} \sim p(\mathbf{v})} \left[ \mathbf{v}^{\mathrm{T}} A \mathbf{v} \right], \quad \mathbf{v} \in \mathbb{R}^d \tag{83}$$

where $p(\mathbf{v})$ is **isotropic**, i.e. $\mathbb{E}_{\mathbf{v} \sim p(\mathbf{v})}[\mathbf{v}\mathbf{v}^T] = I$. Therefore, the trace can be estimated by Monte Carlo:

$$\mathrm{tr}(A) \approx \frac{1}{V} \sum_{i=1}^{V} \mathbf{v}_i^{\mathrm{T}} A \mathbf{v}_i, \tag{84}$$

where each $\mathbf{v}_i \in \mathbb{R}^d$ are $i.i.d.$ samples from $p(\mathbf{v})$.

There are several viable choices for the distribution $p(\mathbf{v})$ in HTE, such as the most common standard normal distribution. Among isotropic distributions, the Rademacher distribution minimizes the variance of HTE. The proof for the minimal variance is given in [36].

Table 12: Performance comparison of STDE-gPINN for high-dimensional inseparable PDEs. "None" in the "gPINN method" column indicates that no gPINN loss was used.

| Equation | gPINN method | Metric | 100 D | 1K D | 10K D | 100K D |
|---|---|---|---|---|---|---|
| Allen-Cahn | JVP-HVP | Speed | 256.75it/s | 249.48it/s | 108.80it/s | 61.04it/s |
| | | Error | 3.97E-02 ±3.98E-04 | 1.02E-03 ±6.89E-05 | 3.08E-04 ±7.48E-06 | 1.39E-03 ±1.42E-05 |
| | STDE | Speed | **366.46it/s** | **324.60it/s** | **207.85it/s** | **155.40it/s** |
| | | Error | 4.34E-02 ±3.72E-04 | 5.26E-04 ±2.26E-05 | 1.25E-03 ±4.07E-05 | 7.61E-04 ±1.03E-04 |
| | None | Error | 4.98E-02 ±3.82E-04 | 6.32E-03 ±4.43E-05 | 1.19E-04 ±1.04E-05 | 5.43E-04 ±4.30E-06 |
| Sine-Gordon | JVP-HVP | Speed | 1008.65it/s | 788.10it/s | 413.32it/s | 107.68it/s |
| | | Error | 1.85E-03 ±4.61E-05 | 1.02E-03 ±6.89E-05 | 1.79E-04 ±1.06E-05 | 5.76E-04 ±1.37E-04 |
| | STDE | Speed | **1165.35it/s** | **948.99it/s** | **542.36it/s** | **210.75it/s** |
| | | Error | 6.69E-03 ±1.48E-04 | 1.12E-03 ±1.38E-05 | 1.76E-04 ±5.31E-06 | 1.55E-03 ±4.30E-05 |
| | None | Error | 4.74E-03 ±6.68E-05 | 7.02E-04 ±1.69E-05 | 1.31E-04 ±1.22E-05 | 8.07E-04 ±4.01E-06 |

## J.2 HTE as the pushforward of dense isotropic random 2-jets

Note that both HTE and the STDE Hessian trace estimator (Eq. ) are computing the quadratic form of Hessian, a specific contraction that is included in the pushforward of 2-jet. In STDE, the random vectors are the unit vectors whose indexes are sampled from the index set without replacement. This can be seen as a discrete distribution $p(\mathbf{v})$ such that $\mathbf{v} = \sqrt{d}\mathbf{e}_i$ for $i = 1, 2, \cdots, d$ with probability $1/d$, which is isotropic. Hence HTE can also be defined as a push forward of random 2-jet that are isotropic.

We can now write the computation of HTE as follows

$$\tilde{\nabla}^2{}_{p,N} u_\theta = \frac{d}{N} \sum_{j=1}^{N} \partial^2 u_\theta(\mathbf{x})(\mathbf{v}_j, \mathbf{0}), \quad \mathbf{v}_j \sim p(\mathbf{v}). \tag{85}$$

where $\tilde{\nabla}^2{}_N$ is the STDE for Laplacian with random jet batch size $N$.

## J.3 Estimating the Biharmonic operator

It was shown in [12] that the Biharmonic operator

$$\Delta^2 u(\mathbf{x}) = \sum_{i=1}^{d} \sum_{j=1}^{d} \frac{\partial^4}{\partial x_i^2 \partial x_j^2} u(\mathbf{x}) \tag{86}$$

has the following unbiased estimator:

$$\Delta^2 u(\mathbf{x}) = \frac{1}{3} \mathbb{E}_{\mathbf{v} \sim p(\mathbf{v})} \left[ \partial^4 u(\mathbf{x})(\mathbf{v}, \mathbf{0}, \mathbf{0}, \mathbf{0}) \right] \tag{87}$$

where $p$ is the $d$-dimensional normal distribution. Therefore its STDE estimator is

$$\tilde{\Delta}^2{}_N u(\mathbf{x}) = \frac{d}{3N} \sum_{j=1}^{N} \partial^4 u(\mathbf{x})(\mathbf{v}_j, \mathbf{0}, \mathbf{0}, \mathbf{0}), \quad \mathbf{v}_j \sim \mathcal{N}(\mathbf{0}, \mathbf{I}) \tag{88}$$

# K STDE with dense jets

## K.1 STDE with second order dense jets as generalization of HTE

Suppose $\mathcal{D}$ is a second-order differential operator with coefficient tensor $\mathbf{C}$. If $\mathbf{C}$ is not symmetric, we can symmetrize it as $\mathbf{C}' = \frac{1}{2}(\mathbf{C} + \mathbf{C}^\top)$, and $D_u^2(\mathbf{a}) \cdot \mathbf{C} = D_u^2(\mathbf{a}) \cdot \mathbf{C}'$ since $D_u^2(\mathbf{a})$ is symmetric.

Furthermore, we can make $\mathbf{C}$ positive-definite by adding a constant diagonal $\lambda \mathbf{I}$ where $-\lambda$ is smaller than the smallest eigenvalue of $\mathbf{C}$. The matrix $\mathbf{C}'' = \frac{1}{2}(\mathbf{C} + \mathbf{C}^\top) + \lambda \mathbf{I}$ then has the eigen decomposition $\mathbf{U}\boldsymbol{\Sigma}\mathbf{U}^\top$ where $\boldsymbol{\Sigma}$ is diagonal and all positive. Now we have

$$\mathbb{E}_{\mathbf{v} \sim \mathcal{N}(\mathbf{0}, \boldsymbol{\Sigma})}[\mathbf{U}\mathbf{v}\mathbf{v}^\top \mathbf{U}^\top] = \mathbf{U}\boldsymbol{\Sigma}\mathbf{U}^\top = \mathbf{C}''. \tag{89}$$

### K.2 Why STDE with dense jets is not generalizable

Specifically, we will prove that it is impossible to construct dense STDE for the fourth-order diagonal operator $\mathcal{L}u = \sum_{i=1}^{d} \frac{\partial^4 u}{\partial x_i^4}$.

The mask tensor of $\mathcal{L}$ is the rank-4 identity tensor $\mathbf{I}_4 \in \mathbb{R}^{d \times d \times d \times d}$, so the condition for unbiasedness is

$$\mathbb{E}_{\mathbf{v} \sim p}[v_i^{(a)} v_j^{(b)} v_k^{(c)} v_l^{(d)}] = M_{ijkl} = \delta_{ijkl}, \quad a, b, c, d \in \{1, 2, 3, 4\} \tag{90}$$

where $\delta_{ijkl} = 1$ when $i = j = k = l$, and is 0 otherwise.

In the most general case where $a \neq b \neq c \neq d$, we can sample $\mathbf{v} \in \mathbb{R}^{4d}$ and split it into four $\mathbb{R}^d$ vectors. In this case we can define blocks of covariance as $\mathbb{E}_{\mathbf{v} \sim p}[\mathbf{v}^{(a)}\mathbf{v}^{(b)}] = \boldsymbol{\Sigma}^{ab}$, and $\boldsymbol{\Sigma} = [\boldsymbol{\Sigma}^{ab}]_{ab}$. Denote the fourth-moment tensor of $p$ as $\mu_{ijkl}$, then Eq. 90 states that the block $\boldsymbol{\mu}^{abcd}$ in the fourth moment tensor should match $\mathbf{C}$. Fourth moments can always be decomposed into second moments:

$$M_{ijkl} = \mu_{ijkl}^{abcd} = \Sigma_{ij}^{ab}\Sigma_{kl}^{cd} + \Sigma_{ik}^{ac}\Sigma_{jl}^{bd} + \Sigma_{il}^{ad}\Sigma_{jk}^{bc} \tag{91}$$

So finding the $p$ that satisfies Eq. 90 is equivalent to finding a zero-mean distribution $p$ with covariance that satisfies the above equation. In the case of $\mathcal{L}$, the mask tensor is block-diagonal: $M_{ijkl} = \sigma_{ij}\delta_{ij,kl}$. So in the case where $a \neq b$, set $a = 1, b = 2$, we have

$$\sigma_{ij} = \mu_{ijij}^{1212} = \Sigma_{ii}^{11}\Sigma_{jj}^{22} + 2(\Sigma_{ij}^{12})^2 \tag{92}$$

and $\boldsymbol{\Sigma} = \begin{bmatrix} \boldsymbol{\Sigma}^{11} & \boldsymbol{\Sigma}^{12} \\ \boldsymbol{\Sigma}^{21} & \boldsymbol{\Sigma}^{22} \end{bmatrix} \in \mathbb{R}^{2d \times 2d}$. Firstly, consider the diagonal entries of $\sigma$:

$$\sigma_{ii} = \mu_{iiii}^{aaaa} = 3(\Sigma_{ii}^{aa})^2, \quad a \in \{1, 2\} \tag{93}$$

This can always be satisfied by setting the diagonal entries of both $\boldsymbol{\Sigma}^{aa}$ and $\boldsymbol{\Sigma}^{aa}$ block as follows:

$$\Sigma_{ii}^{aa} = \sqrt{\sigma_{ii}/3}, \quad a \in \{1, 2\} \tag{94}$$

Next, consider the entire $\sigma$ matrix. We have

$$\sigma_{ij} = \mu_{ijij}^{1212} = \Sigma_{ii}^{11}\Sigma_{jj}^{22} + 2(\Sigma_{ij}^{12})^2 = \frac{1}{3}\sqrt{\sigma_{ii}\sigma_{jj}} + 2(\Sigma_{ij}^{12})^2 \tag{95}$$

In the case of $\mathcal{L}$, we have $\sigma_{ij} = \delta_{ij}$, so for $i \neq j$ we have

$$0 = \frac{1}{3} + 2(\Sigma_{ij}^{12})^2 \tag{96}$$

which is impossible to satisfy since entries in a covariance matrix must be real.

### K.3 Sparse vs dense jets

The variance of the sparse STDE estimator comes from the variance of selected derivative tensor elements, whereas the variance of the dense estimator comes from the derivative tensor elements that are not selected. For example, in the case of Laplacian, as also discussed in [12], the variance of the sparse STDE estimator comes from the diagonal element of the Hessian, whereas the variance of the dense STDE estimator comes from all the off-diagonal element of the Hessian.

## L  Further ablation study

Figure 5: Ablation on randomization batch size with *Inseparable and effectively high-dimensional PDEs*, dim=100k, 5 runs with different random seeds. Model converges when the difference of L2 error is below 1e-7.

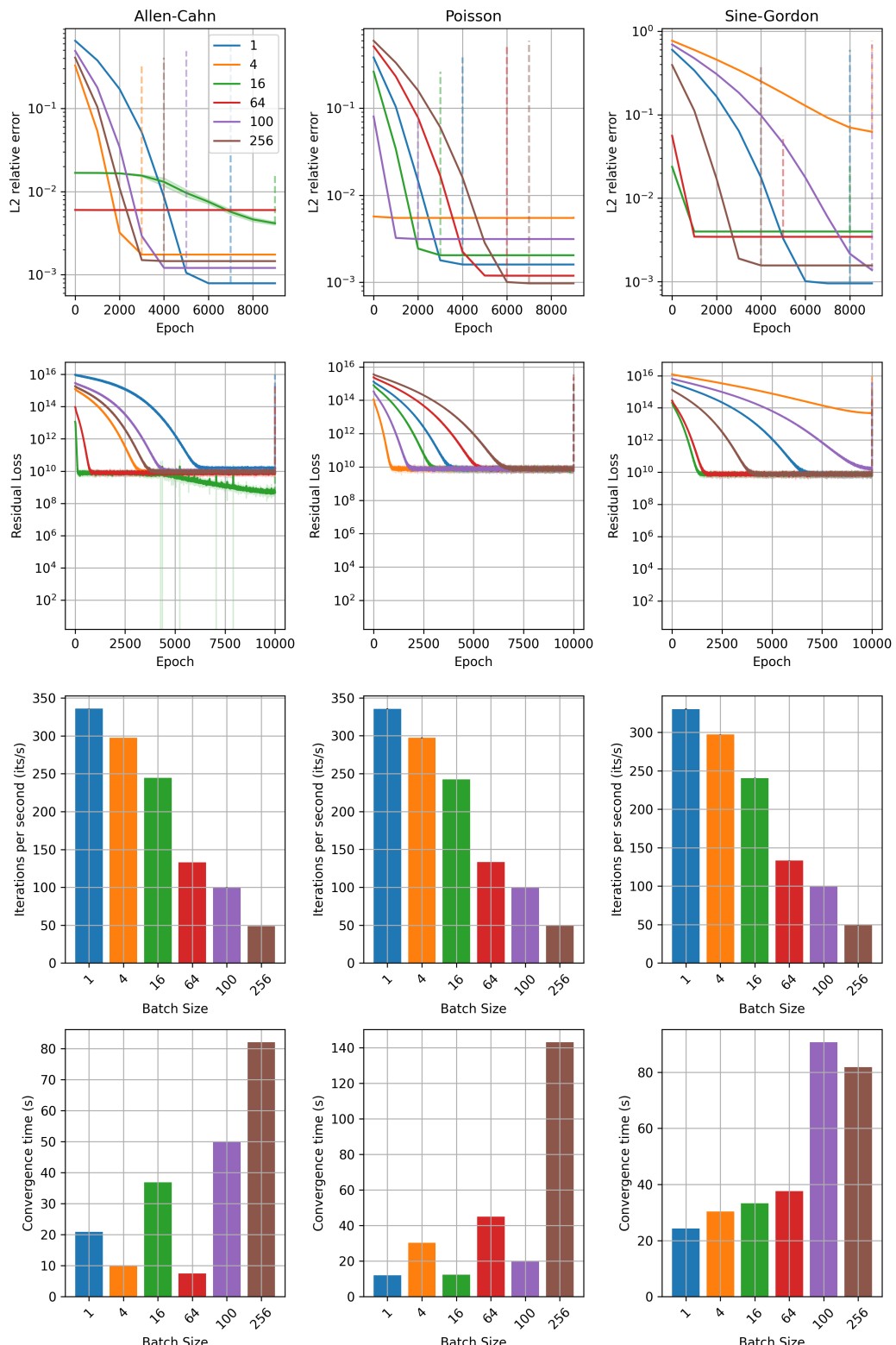

