# OpenReview forum: "Stochastic Taylor Derivative Estimator: Efficient amortization for arbitrary differential operators"
_NeurIPS.cc/2024/Conference — NeurIPS 2024 oral_

### Official Review · Reviewer_cUDy · 2024-06-22

**Soundness:** 3
**Presentation:** 3
**Contribution:** 3
**Rating:** 7
**Confidence:** 4

**Summary:**

The paper proposes a new method, STDE, to compute high-order differential operators of (high-dimensional) neural network-represented functions. Two key ingredients include a generalization of Taylor-mode AD and randomness in the algorithm. This work shows impressive performance of STDE, in terms of both speed and memory.

**Strengths:**

-	This work addresses an important and challenging task in physics-informed machine learning.
-	The illustration and analysis of previous AD algorithms and STDE are clear and comprehensive.
-	STDE displays noticeable potential to improve the computation of arbitrary differential operators in both memory consumption and efficiency.
-	The work provides new insight into introducing randomness other than SDGD or HTE whose basic ideas are employing Monte Carlo over dimensionality.

**Weaknesses:**

1.	Compared with original back-propagation or forward-Laplacian(FL), STDE is a random algorithm that does not give **precise** derivatives. In precision-demanding tasks, e.g. solving many-electron Schrödinger equations, the inaccurate value of differential operators will possibly result in an unreliable solution. To make the result more convincing, a cost-accuracy curve (STDE with different number of random samplings, i.e. $|J|$ in Eqn.(17)) should be added.
Furthermore, in light of this, the comparison between STDE and FL in super high dimensions (in table 1,2) is unfair. A possible remedy is to compare STDE with modified FL, which employs randomness to be more scalable, for instance, Monte Carlo over dimensions. The author can refer to [1] for more technical and implementation details of FL.
2.	All the test cases of high-dimensional high-order operators are Laplacians (including $\Delta^2$). The high-order diagonal differential operator is mentioned in L216 but does not appear in the experiments. A comparison between STDE and direct back-propagation is expected.
The applications are all coming from the small area of PINN. To demonstrate the broad impact of this work, the author should target more tasks of general interest, e.g. second order derivative of NN parameter, which plays an important role in accelerating optimizations.
3.	Some of the highly related existing works are missing. E.g.,
-	Regarding L63, the forward rule for general different operators is done in [1].
-	Regarding the geometric interpretation of Taylor-mode AD in sec 3.4, although not explicitly mentioned in Ref. [6] in the paper, is discussed in section 2 of [2]. The author should cite [6] or the first paper presenting similar ideas, and clarify their novelty if they still hope to claim generalizing univariate Taylor mode AD as their contribution.
4.	As is discussed in Section 4.4, STDE can not be applied to *arbitrary* high-order operators. Obviously, this restricts the potential application scenarios of STDE.
5.	The author didn’t discuss how large $l$ would be for large $k$ (notations from L165), which is necessary to assess the effectiveness of STDE for general high-order derivatives.
6.	Some parts of the writing are confusing. See **Questions**.
Also, there are several misuses of terminologies. E.g.,
-	Fréchet derivative, not Frechet.
-	The form of second order PDE in Eq.(18) is not ‘general’. For instance, you are assuming ellipticity in the second-order term. Please refer to Eq.(10) or Eq.(1) in [1] for *general* second-order operator.
-	In Appendix I.1 and the main text, ‘inseparable and effectively high-dimensional’, I guess what the authors intend to say is ‘elliptic’. The linear part of all the test equations is merely a Laplacian, casting doubt on the effectiveness of STDE for general elliptic equations.
-	In Appendix I.2 and the main text, these PDEs are not nonlinear Fokker-Planck Equations, I think the correct name would be ‘semilinear parabolic equation’.



[1] Li R, Wang C, Ye H, He D, Wang L. DOF: Accelerating High-order Differential Operators with Forward Propagation. ICLR 2024 Workshop on AI4Differential Equations In Science
[2]Tan, Songchen. Higher-Order Automatic Differentiation and Its Applications. Diss. Massachusetts Institute of Technology, 2023.

**Questions:**

1.	Which function does ‘univariate’ refer to in Sec 3.4, $g$ or $F$? What is the first contribution (in the introduction), generalizing Taylor mode AD to the case where $g$ is multivariate, or $F$? I am skeptical of the authors’ claiming it as a contribution/novelty because it seems like this is merely about replacing scalar Faà di Bruno's formula with that of tensors.
2.	In Appendix D, where $g$ defines an $n$-dimensional manifold, is $n$ always set to 1 in this work? If not, please point me to where the method with general $n$ is discussed.
3.	Although the authors claim to remove the exponential complexity w.r.t. the order of derivatives, for general operators the size of coefficient tensor **${C}$** in Eqn(10) grows exponentially and the sampling procedure in Eqn(15) will become hard.
4.	Could the authors think of any real application scenarios with high-order high-dim operators? As for the case of low-dim high-order operators, the authors need to check if the differential operator takes up a considerable proportion of the whole process, e.g. training PINN. If not, the benefit of accelerating differential operators might be very limited.
5.	Does STDE have the potential to make use of sparsity in the neural network function, as shown in FL and [1]? For the specific case of second-order operators, how would STDE perform when $C$ is dense but low-rank?
6.	Instead of tackling a scalar function (e.g. Laplacian of a function), could STDE help with the computation of tensor-value functions, e.g. Hessian or curvature? Hessian appears in Monge-Ampère equation and Ricci curvature appears in Einstein equation.

On writing and notations:
1. Can you clarify on ‘$[2]$’ in L214 with more details or concrete examples? For me, $d^2u_\theta$ in Eqn.(16) is a (0,2)-tensor (Hessian), but why does it take three input $(a,e_j,0)$? Please also clarify on ‘$[3]$’ in Eqn(44).
2.	In Eqn.(19), it should be $\sum_{i=1}^d$.
3.	In L247, do you mean $v\sim p$,, instead of $(a,v,0)\sim p$?
4.	In the ‘100D’ column of table 2, 539 is not the smallest number. Why is it in bold?
5.	There needs to be more description in section F.1.
It is good to remind the readers that Einstein notation is employed in Eq.(45) and that the indices $n,\ n’,\ n’’$ are omitted.
The author should inform the readers that in the discussion $n=1$ for $n$ in Eq(40) and notations like $\partial^4u(x)(e_i,e_j,0,0)$ refers to setting $v^{(1)}=e_i,\ v^{(2)}=e_j$ instead of the taking $e_i,e_j,0,0$ as inputs for a (0-4) tensor.

Overall, this paper has the potential to be a good work. Hope the concerns above can be well addressed and then I will upgrade my assessment accordingly.

**Limitations:**

Yes.

---

> ### Author Rebuttal · Authors · 2024-08-05
>
> **Weakness**
> > 1. Compared with original ...
>
> Thanks for your insightful comment. We want to first clarify that, the main takeaway we wish to show out of this comparison between FL and STDE is to highlight the fact that randomization is important for scalability in dimensionality. While FL removes the redundancy in normal AD, being an exact method its complexity still grows with the dimensionality of the domain. We are not claiming that STDE beats FL, but rather to emphasize that randomization is key and there is potentially a lot of research to be done at the intersection of randomized linear algebra and automatic differentiation. In fact, from Table 2 one can see that in the low-dimensional regime, FL is clearly the best method both in terms of speed and memory.
>
> It is certainly true that a randomized FL will perform much better in terms of scalability, but this entails non-trivial work. In general, for each specific differential operator, there could exist optimized forward rules like FL, as also pointed out by reviewer xY8N. We believe that developing a comprehensive set of such rules is a promising direction, but we think it is out of the scope of this paper.
>
> Regarding the accuracy reduction due to randomization, we would suggest the reviewer inspect the relative error data shown in Table 3. From the table, one can see that L2 error under randomization has the same order as exact methods like FL. However, to ascertain the stochastic error for solving a specific problem like the many-electron Schrödinger equations requires further investigation.
>
> For the cost-accuracy curve you've mentioned, we have added additional experiments where the results are included in the rebuttal PDF. Please refer to the global rebuttal and the PDF attached.
>
> > 2. All the test ...
>
> - Besides the Laplacians and the Biharmonic operator, we also provide an example of amortized g-PINN in Appendix I.4.2 which is both high-dimensional and high-order. Furthermore, we have provided a general procedure for constructing sparse STDE for arbitrary derivative operators in Appendix F.
> - The high-order diagonal differential operator is a straightforward extension of the Laplacian operator.
> - We believe that the comparison between randomized AD and direct back-propagation is redundant as it is already done in previous work ([13]).
> - This work applies to any problem where one needs to compute a differentiable operator on the input of an NN, as explained in the introduction. STDE applies naturally to PINN but it can also be applied to other problems that require input gradients. For example, adversarial attacks, feature attribution, and meta-learning, to name a few. The method is not suited for computing the derivative of NN parameter as explained in Section 3.
>
> > 3. Some of the highly related existing works are missing...
>
> Thank you for pointing out these relevant works that we are not aware of.
> - In [1], an exact AD method is proposed for computing arbitrary **second-order** differential operators. The idea is similar to our idea in equations (19) and (22) where matrix decomposition and change of basis are employed, but the overall algorithm is different. It is worth stressing again that our method applies to **arbitrary derivative order**. We will cite this paper in our final version since it is highly relevant.
> - Regarding section 3.4, this is just background material as it falls under the preliminaries section. We provided a detailed write-up here because it serves as an important context for our main contribution in section 4.1, and we want to set up the notation in a way that facilitates the explanation of our idea. It is true that [2] also discussed the geometric interpretation of Taylor-mode AD, but this is also discussed in earlier literature we cited in Section 2. Nevertheless, we will include [2] in related works in the final version for completeness' sake.
>
> > 4. As is discussed in Section 4.4...
>
> Section 4.4 section refers to the **dense** version of STDE which has limited applicability. The **sparse** STDE is universally applicable. Reviewer yjY8 also asked a question about the comparison between the two, please refer to my answer to that question for further clarification.
>
> > 5. The author didn’t ...
>
> Thanks for bringing up this important question. How large $l$ should be depends on how off-diagonal the operator is. If the operator is diagonal as in the case of equation (17), $l=k$ is enough. If the operator is maximally non-diagonal, i.e. it is a partial derivative where all dimensions to be differentiated are distinct, then the minimum $l$ needed is $(1+k)k/2$. For more details, please refer to Section F where a general procedure for determining the jet structure is discussed.
>
> > 6. Some parts of the writing are confusing...
>
> Thanks for your detailed checking on the correctness of the terminology.
> - When we say equation (18) is 'general' we mean that it includes a large class of commonly used second-order PDEs. We also mention that they are parabolic in the paragraph.
> - When we say 'inseparable and effectively high-dimensional', we are stressing the fact that the solution does not have strong symmetries such that the effective dimension can be drastically reduced via some change of variables. For example, if the solution has spherical symmetry, then the equation becomes a 1D under spherical coordinates.
> - For the equations in Appendix I.2, you are right that they are semilinear parabolic equations. However, they are also specifically driftless Fokker-Planck equations with semilinear extension, and Fokker-Planck equations are a subset of parabolic equations. Similarly, semilinear equations are nonlinear, but there are other types of nonlinearity like quasilinear. So I guess both terminologies are okay, and this is really just a personal preference.
>
> **Questions**
>
> Thanks for these insightful questions. See our response to these questions in the global rebuttal.

---

> > ### Comment · Reviewer_cUDy · 2024-08-09
> >
> > Thanks for your reply.
> >
> > Regarding the ablation study on randomization batchsize you added, I don’t think it is very appropriate since there is also randomness in the optimization. I would instead suggest directly comparing the accuracy on differential operator itself, e.g. Laplacian of a complicated network function. Additionally, do you have any intuition why batchsize=16&64 get stuck whilst both a smaller and a larger batchsize do not?
> >
> > I would suggest adding the clarifications you replied in the final version. Also, I want to comment on some notations and terminologies and do hope the author to make them precise given the diverse background of NeurIPS attendees.
> >
> > >  However, they are also specifically driftless Fokker-Planck equations with semilinear extension.
> >
> > Fokker-Planck equation has a clear definition and derivation from evolution of probability densities. I could not agree with the usage of the terminology here and do think every word should be made precise (even though it is not relevant to the contribution of this work at all.)
> >
> > > I think these two are synonymous, as a here is $a$ not a R.V.
> >
> > Then you should use either $v\sim p$ or $(a,v,0)\sim \delta_a\times p \times \delta$ (or something like that).

---

> > > ### Author Response · Authors · 2024-08-11
> > >
> > > > Regarding the ablation study on randomization batchsize you added, I don’t think it is very appropriate since there is also randomness in the optimization.
> > >
> > > Thank you for your comment on the additional experiment. Firstly we would like to emphasize that, one of the main claims of our paper is when solving optimization problems like equation (1), the cost of computing expensive derivative operators can be amortized over the stochastic optimization of the neural network ansatz. This is why we conduct all the experiments under a stochastic optimization. We regret that this point was not communicated more clearly, and we will try to make this point more prominent in the final version of the paper.
> > >
> > > It is possible to conduct yet another experiment that directly compares the accuracy of differential operators under different randomization batch sizes, and we do expect to see that variance decreases proportional to the randomization batch size. However, this does not determine the effectiveness of STDE as explained above, and we think it would be more interesting to show the effect of randomization batch size on the convergence of various PDEs.
> > >
> > > > Additionally, do you have any intuition why batchsize=16&64 get stuck whilst both a smaller and a larger batchsize do not?
> > >
> > > This was puzzling for us as well. Upon conducting further experiments, we found that it is most likely since we had chosen a set of particularly bad random seeds ({1,2,3,4,5}). We are rerunning the experiments and will report the new result once it is ready.
> > >
> > > > I would suggest adding the clarifications you replied in the final version.
> > >
> > > Yes, we will be revising the final version to incorporate the clarifications we made in the above response. We would like to thank the reviewer again for his/her detailed checking on the terminologies, which greatly improved the quality of our paper.
> > >
> > > > Fokker-Planck equation has a clear definition and derivation from evolution of probability densities. I could not agree with the usage of the terminology here and do think every word should be made precise (even though it is not relevant to the contribution of this work at all.)
> > >
> > > Upon further consideration, we think that it would indeed be more precise to rename nonlinear Fokker-Planck in Appendix I.2 to semilinear parabolic equations, considering the interpretation of the Fokker-Planck equation. Initially, we used the name Fokker-Planck to be consistent with the baseline method SDGD (Section 5.2 in [1]). Thanks again for checking the terminologies used in our paper!
> > >
> > > > Then you should use either $v\sim p$ or $\delta_{a}\times p\times \delta$ (or something like that).
> > >
> > > Thank you for your suggestion. We think both of the suggested modifications are more precise than the current formula used in the paper. We will incorporate this change into the final version of our paper.

---

> > > > ### Comment · Reviewer_cUDy · 2024-08-11
> > > >
> > > > Thanks. I have updated my score.

---

### Official Review · Reviewer_SkVJ · 2024-06-30

**Soundness:** 4
**Presentation:** 3
**Contribution:** 4
**Rating:** 7
**Confidence:** 4

**Summary:**

This paper proposes a stochastic optimization approach to find the minimizer of a cost function, which involves the complicated differential operators. The problem is computationally complex, hence, the authors propose to deal with a minibatch of derivatives in each iteration which reduces computational complexity. The method has application in various learning problems involving complicated differential operators such as physics-informed neural networks. Experiments are provided to evaluate the method.

**Strengths:**

The paper seems to address a difficult optimization problem. The dimension reduction idea seems to be new and interesting. Application to PINN seems also novel.

**Weaknesses:**

NA

**Questions:**

NA

---

> ### Author Rebuttal · Authors · 2024-08-05
>
> Thanks for your positive review of our paper. We would appreciate it if you could provide any suggestions on potential improvements to our paper, and we welcome any future questions you might have on our paper.

---

> ### Comment · Reviewer_SkVJ · 2024-08-11
> **I will keep my rating unless a major weakness is found by other reviewers**
>
> Thanks.

---

### Official Review · Reviewer_yjY8 · 2024-07-12

**Soundness:** 3
**Presentation:** 2
**Contribution:** 3
**Rating:** 7
**Confidence:** 3

**Summary:**

This work proposes a scalable method for the optimization of loss functions including higher-order derivatives. The proposed method interprets arbitrary differential operators as derivative tensor contractions which are then estimated through random contractions. These random contractions can be computed efficiently using Taylor mode AD.

**Strengths:**

1. The proposed method is novel and interesting. I think the idea of random contractions of the derivative tensor is very intuitive and at least in principle should be scalable and a clear improvement over SDGD by ameliorating the exponential scaling in the order of the the differential operator.

2. The idea is technically sound. There is sufficient empirical validation.

2. The well-written presentation of the background material and JAX implementations of SDGD are also useful secondary contributions.

**Weaknesses:**

1. The presentation could be improved a tad bit. I think having 4 pages of background material is a bit unnecessary specially the general background on automatic differentiation could easily be moved to the appendix and space could be better utilized by explaining the method in more detail and discussing the experimental details. For example, a schematic diagram similar to Figure 2 illustrating STDE could be helpful in understanding the proposed method.

**Questions:**

1. In 291 it is stated "Since the only change is how the derivatives are computed, the relative L2 error is expected to be of the same order among different methods". Is this accurate? SDGD and STDE are stochastic approximations whereas Forward Laplace is not.
2. I am a bit unclear about how to choose a distribution over the l-jets that satisfy the unbiasedness condition for any specific problem. It seems that the sparse random jets have the advantage of universally applicable but they also seem to involve a lot of redundant computations.

**Limitations:**

Some limitations are discussed in section 6.

---

> ### Author Rebuttal · Authors · 2024-08-05
>
> **Weaknesses**
> > 1. The presentation could be improved a tad bit. I think having 4 pages of background material is a bit unnecessary specially the general background on automatic differentiation could easily be moved to the appendix and space could be better utilized by explaining the method in more detail and discussing the experimental details. For example, a schematic diagram similar to Figure 2 illustrating STDE could be helpful in understanding the proposed method.
>
> We appreciate your input on writing. We agree that by moving the background section to the appendix we would have more space for experiment details, but doing this could also disturb the flow of the paper, as we discussed the motivation for our method when going through the background material.
>
> **Questions**
> > 1. In 291 it is stated "Since the only change is how the derivatives are computed, the relative L2 error is expected to be of the same order among different methods". Is this accurate? SDGD and STDE are stochastic approximations whereas Forward Laplace is not.
>
> Thanks for pointing out this issue. Yes, you are right, Forward Laplacian is an exact method, so it is expected to perform better in terms of L2 error. However, as we can see in Table 4, the L2 error is of the same order, at least in the case where the dimension is >1k. We will update the description of this section in the final version to reflect this point.
>
> > 2. I am a bit unclear about how to choose a distribution over the l-jets that satisfy the unbiasedness condition for any specific problem. It seems that the sparse random jets have the advantage of universally applicable but they also seem to involve a lot of redundant computations.
>
> Regarding computation cost, it is worth pointing out that both the sparse and the dense versions of STDE would have similar computation costs if the batch size of random jets were the same. The main differences between the sparse and the dense version of STDE are (1) sparse STDE is universally application whereas the dense STDE can only be applied to certain operators, as you've pointed out; (2) the source of variance is different (see Appendix K.3). In general, we would suggest to use sparse STDE unless we know a priori that the sparse version would suffer from excess variance and the dense STDE is applicable.
>
> For constructing STDE for a specific problem, you could refer to the various examples provided in the paper, or follow the method outlined in Appendix F for sparse jet, or Appendix K.1 for dense jet.

---

> > ### Comment · Reviewer_yjY8 · 2024-08-09
> > **Keep Score**
> >
> > Thanks for the clarification! Overall I like the paper and intend to keep the score.

---

### Official Review · Reviewer_xY8N · 2024-07-24

**Soundness:** 4
**Presentation:** 4
**Contribution:** 4
**Rating:** 8
**Confidence:** 5

**Summary:**

The paper addresses the computational challenges of optimizing neural networks with loss functions that include high-dimensional and high-order differential operators. These challenges arise due to the scaling of the derivative tensor size with the dimension of the domain (d) and the computational graph's size with the number of operations (L) and the order of the derivative (k). Traditional methods either amortize the computational cost over the optimization process via randomization or use high-order auto-differentiation (AD) for univariate functions to tackle these issues.

In this work, the authors propose a method to efficiently perform arbitrary contraction of the derivative tensor for multivariate functions. This is achieved by constructing input tangents to univariate high-order AD, enabling efficient randomization of any differential operator. When applied to Physics-Informed Neural Networks (PINNs), this approach provides significant speed and memory efficiency improvements, achieving over 1000 times speed-up and 30 times memory reduction compared to randomization with first-order AD. The method allows solving 1-million-dimensional partial differential equations (PDEs) in just 8 minutes on a single NVIDIA A100 GPU, opening up the possibility of using high-order differential operators in large-scale problems.

**Strengths:**

The paper introduces STDE, a general method for constructing stochastic estimators for arbitrary differential operators, which can be efficiently evaluated using Taylor mode auto-differentiation (AD). When evaluated on Physics-Informed Neural Networks (PINNs), a specific optimization problem where the loss function includes differential operators, STDE significantly outperforms baseline methods. Furthermore, STDE's applicability extends beyond PINNs to arbitrarily high-order and high-dimensional AD-based PDE solvers, making it more general than related methods.

The strengths of this paper are as follows:

Generality: STDE can be applied to a wide range of problems, including those involving arbitrarily high-order and high-dimensional differential operators. This broad applicability distinguishes STDE from other methods, which are often restricted to specific forms of second-order PDEs.

Efficiency: The method enables efficient evaluation of stochastic estimators through Taylor mode AD, providing significant computational benefits.

Performance: In practical evaluations on PINNs, STDE outperforms baseline methods in terms of both speed and memory efficiency. It demonstrates over 1000 times speed-up and 30 times memory reduction compared to first-order AD randomization.

Scalability: STDE allows for the solution of extremely large-scale problems, such as 1-million-dimensional PDEs, in a matter of minutes on advanced hardware like the NVIDIA A100 GPU.

Versatility: Beyond PINNs, STDE can be applied to various AD-based PDE solvers, making it a versatile tool for tackling a broad spectrum of differential operator-based optimization problems.

Overall, STDE's generality, efficiency, performance, scalability, and versatility make it a powerful method for addressing high-dimensional and high-order differential operator challenges in neural network optimization.

**Weaknesses:**

While the paper demonstrates the significant strengths and broad applicability of STDE, it is important to acknowledge some limitations and areas for future improvement. As a general method, STDE may not leverage the specific optimization possibilities that are available for particular operators. Additionally, the study did not explore variance reduction techniques, which could potentially enhance the method's performance and could be a promising area for future research.

Another observation is that while reducing the randomization batch size improves both the speed and memory profile of STDE, this comes with a trade-off in the form of increased computational variance. Further analysis is required to understand and optimize this balance between computational efficiency and variance.

Looking ahead, the paper identifies an intriguing connection between the fields of automatic differentiation (AD) and randomized numerical linear algebra, highlighting the potential for future work at this intersection. Such research could lead to significant advancements in large-scale scientific modeling with neural networks.

In summary, while there are areas for refinement, the contributions of this paper are substantial. The development of STDE as a general and efficient method for constructing stochastic estimators for arbitrary differential operators is a notable achievement, offering substantial benefits for high-dimensional and high-order differential operator problems in neural network optimization. The identified limitations and future research directions provide a clear path for further enhancing this already impressive work.

**Questions:**

Given the importance of various other complex equations in scientific modeling, I am curious about the applicability of STDE to equations such as the Nonlinear Schrödinger Equation (NLS), the fourth-order NLS, and the Navier-Stokes equations. Could you elaborate on how STDE might perform or be adapted for these specific cases? Additionally, are there any preliminary results or theoretical considerations you could share regarding the application of STDE to these important equations?

**Limitations:**

Yes, they have.

---

> ### Author Rebuttal · Authors · 2024-08-05
>
> **Weaknesses**
> > While the paper demonstrates the significant strengths and broad applicability of STDE, it is important to acknowledge some limitations and areas for future improvement. As a general method, STDE may not leverage the specific optimization possibilities that are available for particular operators. Additionally, the study did not explore variance reduction techniques, which could potentially enhance the method's performance and could be a promising area for future research.
>
> Thanks for your suggestion on future improvement. You are right that there could exist a more efficient scheme tailored for specific operators like the Laplacian. However, to derive and implement a comprehensive set of optimized derivative estimators is a challenging task whose workload is akin to implementing an AD framework, therefore out of the scope of the current paper. Nevertheless, we hope that our paper can inspire future work in this direction. The same goes for variance reduction. We have done some preliminary work on variance reduction which will form a separate paper from this one.
>
> > Another observation is that while reducing the randomization batch size improves both the speed and memory profile of STDE, this comes with a trade-off in the form of increased computational variance. Further analysis is required to understand and optimize this balance between computational efficiency and variance.
>
> Thanks for pointing out this. We have conducted further ablation studies on the randomization batch size, please refer to the global rebuttal and the PDF attached.
>
> **Questions**
> > Given the importance of various other complex equations in scientific modeling, I am curious about the applicability of STDE to equations such as the Nonlinear Schrödinger Equation (NLS), the fourth-order NLS, and the Navier-Stokes equations. Could you elaborate on how STDE might perform or be adapted for these specific cases? Additionally, are there any preliminary results or theoretical considerations you could share regarding the application of STDE to these important equations?
>
> For the NLS, the nonlinear term $|\psi |^2\psi$ does not contain derivatives, and the Laplacian term can be handled by STDE similar to the experiments we have done in this paper. As for the fourth-order NLS, the fourth-order term is a Biharmonic operator which can be handled by STDE as well as discussed in Appendix J.3, as well as in [12]. As for the Navier-Stokes equation, it is both low-dimensional and low-order, so STDE may not provide any significant acceleration. However, if we consider generalized Navier-Stokes equations that include higher-order spatial derivatives to account for dispersive effects, similar to generalized KdV equations discussed in Appendix I.4.1, then STDE might provide some acceleration.

---

> > ### Comment · Reviewer_xY8N · 2024-08-09
> >
> > Thank you very much for your answer. I will keep my score.

---

### Author Rebuttal · Authors · 2024-08-05

**Additional Experiments**

We have conducted further ablation studies on the randomization batch size (see the attached PDF). We ran all three equations from the Inseparable and effectively high-dimensional PDEs (Appendix I.1) with moderately high dimensions (100k), with different randomization batch sizes (1, 4, 16, 64, 100, 256). And we compared their final L2 error and convergence time. As expected, with a smaller batch size the iterations per second are higher (third row of the figure). Memory cost remains roughly the same since the computation graph was not changed. Rather surprisingly, the final L2 error and the time for convergence did not exhibit a linear relationship to randomization batch size, as can be seen in the first row and last row of the figure. Specifically, one can see that the batch size of 1 provides good L2 error and convergence time, regardless of the equation chosen. One explanation is that stochastic optimizers like Adam already have built-in mechanisms to control the variance and are robust to noise during training, so smaller batch sizes can perform well. This warrants further investigation that is out of the scope of this paper.

**Questions from reviewer cUDy**

Reviewer cUDy asked a lot of insightful questions. We answer them here as it would benefit all reviewers to better understand our papers.

> 1. Which function does 'univariate' refer to ...

Firstly we would like to clarify that, our contribution is the insight that the scalar version of Faà di Bruno's formula can be used to perform derivative tensor contractions for multivariate functions, and the vector-valued version of Faà di Bruno's formula is not needed, as discussed in section 4.1. Although the vector-valued Faà di Bruno's formula is introduced in Appendix D.2 (equation 41), it is **not** used in our method. We only use the univariate version (equation 42). In fact, one of our main insights was that for multivariate functions, the scalar version of Faà di Bruno's formula can be used to perform derivative tensor contractions,

As for your other question, in section 4.1, the multivariate function that we perform tensor contraction on is $F$. In section 3.4, the univariate function we are referring to is $F\circ g$, where the high-order chain rule is given by the scalar version of Faà di Bruno's formula.

> 2. In Appendix D ...

In Appendix D we discussed both the scalar and the vector version of Faà di Bruno's formula. The scalar version corresponds to the case of $n=1$ and the vector version corresponds to the case of $n>1$. In this work, we only consider $n=1$, which is enough for our method as discussed in section 4.1.

> 3. Although the authors claim...

Typically we would assume that the coefficient tensor has a certain structure. For example, in the case of equation 77, the operator $\frac{\partial^{3}}{\partial x_{j} \partial x_{i}^{2}}u$ is the diagonal of every rank-2 slice over the first axis of $D^3_u$. So one could sample the index $jii$ by sampling $j$ and $i$ separately from $[1,N]$. Even in the most general case where the index set to be sampled has no apparent structure, sampling the index set is still cheaper than computing the whole derivative tensor.

> 4. Could the authors ...

One example would be the many-body Schrödinger equations, where we need to compute a high-dimensional Laplacian. Another example is the high-dimensional Black-Scholes equation, which has numerous uses in mathematical finance.

> 5. Does STDE ..

In FL and [1], sparsity of intermediate derivative tensor are exploited to save memory and compute. The same idea can be applied to STDE, but it would require some modification to the JAX Taylor mode library. We have not yet done this, but it is an interesting future direction to work on.

> 6. Instead of tackling ...

In general, STDE provides a way to compute high-order derivative tensor elements with one forward pass. For high-order and low-dimensional operators, if the derivative tensor is sparse, then acceleration is possible, as shown by the experiments in section I.4.1.

In the case of Monge-Ampère equation, there is no sparsity since the operator $\det(\text{Hess} u)$ contains all entries from the second-order derivative tensor (Hessian matrix). One still might perform sampling among the additive terms in the determinant, which could provide acceleration when the dimension is high.

In the case of Ricci curvature, the operator is highly nonlinear since it involves computing the inverse of the metric tensor. In this case, STDE cannot be applied straightforwardly since obtaining an unbiased estimator usually requires linearity.

> On writing and notations:

> 1. Can you clarify ...

$d^2u_\theta$ does not denote the Hessian, but rather the second-order pushforward as described in section 3.4, equation 7. The input $(a,e_j,0)$ is a 2-jet, where $a$ is the primal, and $e_j$ and $0$ are the tangents.

The need for indexing notation like $[2]$ comes from the fact that the output is also a jet, which is a tuple that contains the primal and the tangents. We use the indexing notation to select a specific output tangent from the output jet.

> 2. In Eqn.(19), it should be $\sum_{i=1}^d$.

Yes, you are right, thanks for pointing out the typo.

> 3. In L247, do you mean $v\sim p$,, instead of $(a,v,0)\sim p$?

I think these two are synonymous, as $a$ here is not a R.V.

> 4. In the ‘100D’ column of table 2, 539 is not the smallest number. Why is it in bold?

Thanks for pointing out this typo, the smallest memory usage should be 507MB achieved by FL.

> 5. There needs to ...

In the Appendix, Einstein notation was first used and mentioned in equation 41.

As for equation 45, we are using the scalar version of Faà di Bruno's formula, so there are no $n,\ n’,\ n’’$ indices.

$\partial^4$ refers to the fourth-order Fréchet derivative, whose definition is given in equation 35. It is not a tensor. The inputs are the tangent vectors.

---

### Decision · Program_Chairs · 2024-09-25

**Decision:**

Accept (oral)

**Comment:**

The authors describe an interesting combination of randomized approximation and auto-differentiation for high-order tensors arising from physics-informed neural network training.  The paper received positive reviews even before rebuttal, and the reviewers are clearly unanimous in their support at the end of the discussion period.  This seems like a clear accept.